# On the Solution of Thermal Buckling Problem of Moderately Thick Laminated Conical Shells Containing Carbon Nanotube Originating Layers

**DOI:** 10.3390/ma15217427

**Published:** 2022-10-23

**Authors:** Mahmure Avey, Nicholas Fantuzzi, Abdullah Sofiyev

**Affiliations:** 1Division of Mathematical Engineering, Graduate School of Istanbul Technical University, Istanbul 34469, Turkey; 2Analytical Information Resources Center, UNEC-Azerbaijan State Economics University, 1001 Baku, Azerbaijan; 3University-Business Relations Application and Research Center, Istanbul Ticaret University, Istanbul 34445, Turkey; 4Department of Civil, Chemical, Environmental, and Materials Engineering, University Bologna, Viale Risorgimento 2, 40136 Bologna, Italy; 5Distance Education Application and Research Center, Istanbul Ticaret University, Istanbul 34445, Turkey; 6Scientific Research Centers for Composition Materials, UNEC-Azerbaijan State Economics University, 1001 Baku, Azerbaijan

**Keywords:** CNT, nanocomposites, laminated truncated conical shell, thermal buckling, transverse shear stress functions, partial differential equations

## Abstract

This study presents the solution for the thermal buckling problem of moderately thick laminated conical shells consisting of carbon nanotube (CNT) originating layers. It is assumed that the laminated truncated-conical shell is subjected to uniform temperature rise. The Donnell-type shell theory is used to derive the governing equations, and the Galerkin method is used to find the expression for the buckling temperature in the framework of shear deformation theories (STs). Different transverse shear stress functions, such as the parabolic transverse shear stress (Par-TSS), cosine-hyperbolic shear stress (Cos-Hyp-TSS), and uniform shear stress (U-TSS) functions are used in the analysis part. After validation of the formulation with respect to the existing literature, several parametric studies are carried out to investigate the influences of CNT patterns, number and arrangement of the layers on the uniform buckling temperature (UBT) using various transverse shear stress functions, and classical shell theory (CT).

## 1. Introduction

Laminated anisotropic shells have been the subject of much research because they are used as the main bearing elements of engineering structures used in modern aerospace and rocket technology, shipbuilding, energy and chemical engineering, and other fields. The widespread use of composites as structural elements, which can best meet the demands of harsh working conditions, has revealed the need for the application of new theories and methods in the mechanics of laminated composites and structural mechanics based on them. These factors have led to the development of various refined theories instead of the classical shell theory, which neglects the effect of transverse shear deformations in the calculation of laminated anisotropic plates and shells. The formation and development of these theories are summarized in monographs published in different periods [1,2,3,4,5,6]. Since laminated homogeneous shells made of traditional composites are used in thermal environments, their thermal buckling behavior has always been the focus of attention for researchers [7,8,9,10,11,12,13]. 

Extreme static and dynamic loading conditions, thermal and chemical environments, radiation exposure, and other fundamental factors increase the demands on the reliability of structural elements in areas where pioneering technology is used, such as space and aerospace. These requirements are trying to be met by new generation composites created by materials scientists. The development of methods for obtaining new generation composite materials has led to the formation of robust composite structural elements with low thermal conductivity that are as light as possible due to the predicted variable geometric properties. Among these materials, a material type that stands out, with its excellent thermal and mechanical properties, is graphene. In particular, the discovery of CNTs has increased their popularity, as well as the availability of nanocomposites. Today, among the nanocomposites formed by the reinforcement of the main matrix with carbon nanotubes, polymer-based nanocomposites are more popular than metal and ceramic-based nanocomposites, and are used more frequently in various industries [14,15,16,17,18,19,20,21,22,23,24,25]. 

Single-layer heterogeneous composites reinforced with carbon nanotubes, which have robust heat resistance and high strength, are frequently used in the spacecraft and aerospace industries because of their good performance in very high temperature conditions, and the number of studies on their thermal buckling behavior is increasing rapidly [26,27,28,29,30,31,32,33,34,35,36,37,38,39,40,41,42]. In the above studies, analytical and computational methods have been developed, and analyses have been carried out, to solve the thermal and thermoelastic buckling problems of single-layer CNT reinforced beams, plates, and shells with different configurations within the framework of different theories. The number of publications on thermal buckling of laminated shells consisting of CNT reinforced layers is considerably scarce [43,44,45,46,47].

As the main elements of structural systems, the laminated conical shells consisting of nanocomposite layers are often subjected to thermal loads. It was interesting to study their behavior during thermal buckling. The sensitive structure of laminated conical shells consisting of CNT reinforced layers complicates the derivation of the fundamental differential equations in the ST framework, and their solution, due to the sufficiently complex coefficients they contain. In addition, the mathematical modeling of thermal loadings, and their mounting on the subject, make the solution more difficult to carry out. The literature review reveals that the thermal buckling of laminated conical shells composed of CNT originating layers has not been addressed. The main purpose of this study is to solve the thermal buckling problem of moderately thick laminated conical shells consisting of CNT originating layers within the framework of shear deformation theory, and to obtain a new analytical expression in its most general form. 

The manuscript is arranged as follows: The material properties of the nanocomposite layers and laminated conical shells reinforced with carbon nanotubes are modeled in Section 2. In Section 3, basic relations and equations are derived. The solution procedure in the framework of ST and CT is given in detail in Section 4. Numerical results and discussions are given in Section 5 (followed by concluding remarks in Section 6).

## 2. Theoretical Development

### 2.1. Formulation of Problem

Consider the moderately thick laminated truncated conical shell consisting of CNT originating layers with the coordinates S,θ,z along the meridional, circumferential and thickness direction, respectively (as presented in Figure 1). The laminated nanocomposite conical shell is subjected to a uniform temperature rise. The total thickness of the laminated truncated conical shell is h, the length is L, the semi-vertex angle is γ, and the radii are $r_{1}$ and $r_{2}$. Let the coordinate system OSθz be chosen, such that the origin *O* is at the vertex of the whole cone. The S axis lies on the curvilinear middle surface of the cone—$a_{1}$ and $a_{2}$ being the coordinates of the points where this axis intersects the small and large bases, respectively. The mid-surface z=0 is located at the interface of the layers for even values of N, while the mid-surface for the odd values of N is located in the mid-surface of the middle lamina (Figure 1b). The displacement components of the mid-surface along the S,θ and z axes are designated by u,v and w, respectively. The mid-surface rotations of the normals about θ and S axes are denoted by $\varphi_{1}$ and $\varphi_{2}$, respectively. The stress resultants are given by ψ, as in [48,49]:(1)$$\left(T_{S},T_{\theta },\;T_{S\theta }\right)=h\left(\frac{1}{S^{2}}\frac{\partial^{2}\psi }{\partial \theta_{1}^{2}}+\frac{1}{S}\frac{\partial \psi }{\partial S},\;\;\;\;\frac{\partial^{2}\psi }{\partial S^{2}},\;\;\;-\frac{1}{S}\frac{\partial^{2}\psi }{\partial S\partial \theta_{1}}+\frac{1}{S^{2}}\frac{\partial \psi }{\partial \theta_{1}}\right)$$
where $\theta_{1}=\theta \;\sin\;\gamma$

### 2.2. Material Properties of Nanocomposite Layers

The effective material properties and thermal expansion coefficients of nanocomposite layer kth are given by [43,44,45,46,47]:(2)$$\begin{array}{l} E_{S}^{\left(k\right)}(Z)=\eta_{1}^{\left(k\right)}V_{cn}^{\left(k\right)}Y_{Scn}^{\left(k\right)}+V_{m}^{\left(k\right)}Y_{m}^{\left(k\right)},\;\;\;E_{\theta }^{\left(k\right)}(Z)=\frac{\eta_{2}^{\left(k\right)}Y_{m}^{\left(k\right)}Y_{\theta cn}^{\left(k\right)}}{Y_{\theta cn}^{\left(k\right)}V_{m}^{\left(k\right)}+Y_{m}^{\left(k\right)}V_{cn}^{\left(k\right)}},\;\;\;G_{S\theta }^{\left(k\right)}(Z)=\frac{\eta_{3}^{\left(k\right)}G_{m}^{\left(k\right)}G_{S\theta cn}^{\left(k\right)}}{G_{S\theta cn}^{\left(k\right)}V_{m}^{\left(k\right)}+G_{m}^{\left(k\right)}V_{cn}^{\left(k\right)}}\\ G_{Sz}^{\left(k\right)}(Z)=G_{S\theta }^{\left(k\right)}(Z),\;\;\;G_{\theta z}^{\left(k\right)}(Z)=1.2G_{S\theta }^{\left(k\right)}(Z),\;\;\;\nu_{S\theta }^{\left(k\right)}=V_{cn}^{*(k)}\nu_{S\theta cn}^{\left(k\right)}+V_{m}^{\left(k\right)}\nu_{m}^{\left(k\right)} \end{array}$$
and
(3)$$\begin{array}{l} \alpha_{S}^{\left(k\right)}(Z)=\frac{\alpha_{Scn}^{\left(k\right)}V_{cn}^{\left(k\right)}(Z)Y_{Scn}^{\left(k\right)}+V_{m}^{\left(k\right)}Y_{m}^{\left(k\right)}\alpha_{m}^{\left(k\right)}}{V_{cn}^{\left(k\right)}(Z)Y_{Scn}^{\left(k\right)}+V_{m}^{\left(k\right)}Y_{m}^{\left(k\right)}},\;\;\;\\ \alpha_{\theta }^{\left(k\right)}(Z)=(1+\nu_{S\theta cn}^{\left(k\right)})\alpha_{\theta cn}^{\left(k\right)}V_{cn}^{\left(k\right)}(Z)+(1+\nu_{m}^{\left(k\right)})V_{m}^{\left(k\right)}\alpha_{m}^{\left(k\right)}-\nu_{S\theta cn}^{\left(k\right)}\alpha_{S}^{\left(k\right)}(Z) \end{array}$$

Here, $\;Z=\frac{z}{h},\;\;\;-\frac{1}{2}+\frac{k-1}{N}\leq Z\leq -\frac{1}{2}+\frac{k}{N},\;\;\;k=1,2,\ldots ,N$, $Y_{m}^{\left(k\right)},\;\;G_{m}^{\left(k\right)},\;\;\nu_{m}^{\left(k\right)},\alpha_{m}^{\left(k\right)}$ are the Young and shear moduli, Poisson’s ratio, the thermal expansion coefficient in the lamina kth, $Y_{Scn}^{\left(k\right)},\;\;Y_{\theta cn}^{\left(k\right)},\;\;G_{S\theta cn}^{\left(k\right)},\;\;\nu_{S\theta cn}^{\left(k\right)},\;\;\alpha_{Scn}^{\left(k\right)},\alpha_{\theta cn}^{\left(k\right)}$ are the similar elastic and thermal properties for CNTs in the lamina kth, and $\eta_{i}^{\left(k\right)}(i=1,2,3)$ is the efficiency parameter for the lamina kth. The following equality is satisfied for the volume fraction of CNTs and lamina: $V_{cn}^{\left(k\right)}+V_{m}^{\left(k\right)}=1$. 

The distribution of volume fractions for CNTs across the thickness of the layer kth is modeled as the U-, V-, O- and X-shaped elements (See, Figure 2): (4)$$V_{cn}^{\left(k\right)}=\left\{ \begin{array}{l} U\;\;\;\;\;\;\;at\;\;\;\;V_{cn}^{*(k)}\\ V\;\;\;\;\;\;at\;\;\;\;2\left(0.5-Z\right)V_{cn}^{*(k)}\\ O\;\;\;\;\;at\;\;\;\;2\left(1-2\left|Z\right|\right)V_{cn}^{*(k)}\\ X\;\;\;\;at\;\;\;\;4\left|Z\right|\;V_{cn}^{*(k)} \end{array}\right.$$
where $V_{cn}^{*(k)}$ is the total volume fractions of CNTS.

The distribution of volume fractions of CNTs across the thickness of the layer are illustrated in Figure 2

## 3. Basic Relations and Equations

In this section, the basic relations and equations of moderately thick laminated orthotropic conical shells consisting of CNT originating layers are reviewed. In the presence of a temperature field, the constitutive relations for the CNT originating layer kth in the framework of STs can be determined by a generalization of Hooke’s law as follows [26,47]: (5)$$\sigma_{S}^{\left(k\right)}=Q_{11Z}^{\left(k\right)}e_{S}\;+Q_{12Z}^{\left(k\right)}e_{\theta }+\sigma_{ST}^{\left(k\right)},\;\sigma_{\theta }^{\left(k\right)}=Q_{21Z}^{\left(k\right)}e_{S}\;+Q_{22\bar{z}}^{\left(k\right)}e_{\theta }+\sigma_{\theta T}^{\left(k\right)},\;\;\sigma_{S\theta }^{\left(k\right)}=Q_{66Z}^{\left(k\right)}\gamma_{S\theta }$$
and
(6)$$\sigma_{Sz}^{\left(k\right)}=Q_{55Z}^{\left(k\right)}\gamma_{Sz},\;\;\;\;\sigma_{\theta z}^{\left(k\right)}=Q_{44Z}^{\left(k\right)}\gamma_{\theta z}\;$$
where $\sigma_{S}^{\left(k\right)},\sigma_{\theta }^{\left(k\right)},\sigma_{S\theta }^{\left(k\right)},\sigma_{Sz}^{\left(k\right)},\sigma_{\theta z}^{\left(k\right)}$ are the stresses in the kth layer, $e_{S},e_{\theta },\gamma_{S\theta },\gamma_{Sz},\gamma_{\theta z}$ are the strains, and $Q_{ij\bar{z}}^{\left(k\right)},\;(i,\;j=1,2,6)$ is the plane stress-reduced stiffnesses defined in terms of engineering constants in the material axes of the lamina kth. It is given by:(7)$$\begin{array}{l} Q_{11Z}^{\left(k\right)}=\frac{E_{S}^{\left(k\right)}(Z)}{1-\nu_{S\theta }^{\left(k\right)}\nu_{\theta S}^{\left(k\right)}},\;\;\;Q_{22Z}^{\left(k\right)}=\frac{E_{\theta }^{\left(k\right)}(Z)}{1-\nu_{S\theta }^{\left(k\right)}\nu_{\theta S}^{\left(k\right)}},\;\;\;Q_{12Z}^{\left(k\right)}=\nu_{\theta S}^{\left(k\right)}Q_{11Z}^{\left(k\right)}=\nu_{S\theta }^{\left(k\right)}Q_{22Z}^{\left(k\right)}=Q_{21Z}^{\left(k\right)},\;\;\\ Q_{44Z}^{\left(k\right)}=G_{\theta }^{\left(k\right)}(Z),\;\;\;Q_{55Z}^{\left(k\right)}=G_{S}^{\left(k\right)}(Z),\;\;Q_{66Z}^{\left(k\right)}=G_{S\theta }^{\left(k\right)}(Z). \end{array}$$
in which $\sigma_{ST}^{\left(k\right)}$ and $\sigma_{\theta T}^{\left(k\right)}$ are thermal stresses, and they are given by
(8)$$\sigma_{ST}^{\left(k\right)}=-\frac{E_{S}^{\left(k\right)}(Z)\alpha_{S}^{\left(k\right)}(Z)\Delta T}{1-\nu_{S\theta }^{\left(k\right)}\nu_{\theta S}^{\left(k\right)}},\;\;\sigma_{\theta T}^{\left(k\right)}=-\frac{E_{\theta }^{\left(k\right)}(Z)\alpha_{\theta }^{\left(k\right)}(Z)\Delta T}{1-\nu_{S\theta }^{\left(k\right)}\nu_{\theta S}^{\left(k\right)}}$$
where $\Delta T=T-T_{0}$ is the uniform temperature rise from the reference temperature ($T_{0}=300K$) at which the cone is free of thermal stresses. 

The stresses $\sigma_{Sz}^{\left(k\right)}$ and $\sigma_{\theta z}^{\left(k\right)}$ in the lamina kth are expressed by $\varphi_{1}$ and $\varphi_{2}$ as follows [1,47]:(9)$$\sigma_{Sz}^{\left(k\right)}=\frac{df_{1}^{\left(k\right)}(z)}{dz}\varphi_{1},\;\;\sigma_{\theta z}^{\left(k\right)}=\frac{df_{2}^{\left(k\right)}(z)}{dz}\varphi_{2}$$
where $f_{i}^{\left(k\right)}(z),(i=1,2)$ is the distribution function of the transverse shear stresses along the thickness of the *k*th layer.

Using (8), (5) and (6), the strains $\left(e_{S},e_{\theta },\gamma_{S\theta }\right)$ can be expressed as those of their mid-surface $\left(e_{0S},\;e_{0\theta }\;\;\gamma_{0S\theta }\right)$ as follows:(10)$$\left[\begin{array}{l} e_{S}\\ e_{\theta }\\ \gamma_{S\theta } \end{array}\right]=\left[\begin{array}{l} e_{0S}-z\frac{\partial^{2}w}{\partial S^{2}}+J_{1}^{\left(k\right)}(z)\frac{\partial \varphi_{1}}{\partial S}\\ e_{0\theta }-\frac{z}{S}\left(\frac{1}{S}\frac{\partial^{2}w}{\partial \theta_{1}^{2}}-\frac{\partial w}{\partial S}\right)+\frac{J_{2}^{\left(k\right)}(z)}{S}\frac{\partial \varphi_{2}}{\partial S}\\ \gamma_{0S\theta }-\frac{2z}{S}\left(\frac{\partial^{2}w}{\partial S\partial \theta_{1}}-\frac{1}{S}\frac{\partial w}{\partial \theta_{1}}\right)+\frac{J_{1}^{\left(k\right)}(z)}{S}\frac{\partial \varphi_{1}}{\partial \theta_{1}}+J_{2}^{\left(k\right)}(z)\frac{\partial \varphi_{2}}{\partial S} \end{array}\right]$$
where $J_{i}^{\left(k\right)}(z)$ is defined as
(11)$$J_{1}^{\left(k\right)}(z)=\int_{0}^{z}\frac{1}{Q_{55}^{\left(k\right)}(Z)}\frac{df_{1}^{\left(k\right)}(z)}{dz}dz;\;\;\;\;\;\;\;J_{2}^{\left(k\right)}=\int_{0}^{z}\frac{1}{Q_{44}^{\left(k\right)}(Z)}\frac{df_{2}^{\left(k\right)}(z)}{dz}dz$$

The in-plane forces ($T_{S},T_{\theta }\;,T_{S\theta }$), moments ($M_{S},M_{\theta }\;,M_{S\theta }$), and transverse shear forces ($Q_{S},Q_{\theta }$) for laminated nanocomposite truncated conical shells composed of CNT originating layers are obtained from the following integrals [1,2,3,4,5,6,48]:(12)$$\begin{array}{l} \left(T_{ij},\;M_{ij}\right)=\sum_{k=1}^{N}\int_{t_{k-1}}^{t_{k}}\left(1,\;z\right)\sigma_{ij}^{\left(k\right)}dz,\;\;\;\;(i,j=S,\theta ),\\ \left(Q_{S},Q_{\theta }\right)=\sum_{k=1}^{N}\int_{t_{k-1}}^{t_{k}}\left(\sigma_{Sz}^{\left(k\right)},\sigma_{\theta z}^{\left(k\right)}\right)dz\; \end{array}$$
where $t_{k-1}=-0.5h+(k-1)hN^{-1}$ and $t_{k}=-0.5h+khN^{-1}$.

The resultants for the thermal forces and moments are [26]:(13)$$\left[\begin{array}{l} T_{S}^{T},\;M_{S}^{T}\\ T_{\theta }^{T},M_{\theta }^{T} \end{array}\right]\;=\sum_{k=1}^{N}\int_{t_{k-1}}^{t_{k}}\left[\begin{array}{l} Q_{11}^{\left(k\right)}(Z),\;Q_{12}^{\left(k\right)}(Z)\\ Q_{21}^{\left(k\right)}(Z),\;Q_{22}^{\left(k\right)}(Z) \end{array}\right]\left[\begin{array}{l} \alpha_{S}^{\left(k\right)}\\ \alpha_{\theta }^{\left(k\right)} \end{array}\right](1,z)\Delta Tdz,$$

The basic equations for a truncated conical shell, based on the STs, are expressed as [48]:(14)$$\begin{array}{l} \frac{\partial M_{S}}{\partial S}+\frac{M_{S}-M_{\theta }}{S}+\frac{1}{S}\frac{\partial M_{S\theta }}{\partial \theta_{1}}-Q_{S}=0\\ \frac{\partial M_{S\theta }}{\partial S}+\frac{1}{S}\frac{\partial M_{\theta }}{\partial \theta_{1}}+\frac{2M_{S\theta }}{S}-Q_{\theta }=0\\ \frac{\cot\gamma }{S}\frac{\partial^{2}w}{\partial S^{2}}-\frac{1}{S}\frac{\partial^{2}\gamma_{0S\theta }}{\partial S\partial \theta_{1}}-\frac{1}{S^{2}}\frac{\partial \gamma_{0S\theta }}{\partial \theta_{1}}+\frac{\partial^{2}e_{0\theta }}{\partial S^{2}}+\frac{1}{S^{2}}\frac{\partial^{2}e_{0S}}{\partial \theta_{1}{}^{2}}+\frac{2}{S}\frac{\partial e_{0\theta }}{\partial S}-\frac{1}{S}\frac{\partial e_{0S}}{\partial S}=0\\ \frac{\partial Q_{S}}{\partial S}+\frac{Q_{S}}{S}+\frac{1}{S}\frac{\partial Q_{\theta }}{\partial \theta_{1}}+\frac{T_{\theta }}{S\tan\gamma }+T_{S}^{0}\frac{\partial^{2}w}{\partial S^{2}}+\frac{T_{\theta }^{0}}{S}\left(\frac{1}{S}\frac{\partial^{2}w}{\partial \theta_{1}{}^{2}}+\frac{\partial w}{\partial S}\right)+2T_{S\theta }^{0}\frac{\partial }{\partial S}\left(\frac{1}{S}\frac{\partial w}{\partial \theta_{1}}\right)=0 \end{array}$$
where $T_{S}^{0},{\;T}_{\theta }^{0}$, and $T_{S\theta }^{0}$ are the membrane forces for the condition with zero initial moments. Since the temperature is constant in the longitudinal and circumferential directions of the laminated conical shell, and varies only in the thickness direction, the prebuckling deformation can be expressed by the following equations:(15)$$u=0,\;\;v=0,\;\;T_{\theta }^{0}=T_{S\theta }^{0}=0$$

Thus, the prebuckling thermal force $T_{S}^{0}$ is defined in [49]:(16)$$T_{S}^{0}=-\Gamma_{T}$$
where $\Gamma_{T}$ is the thermal parameter. When the temperature changes uniformly throughout the thickness of laminated nanocomposite conical shells, the thermal parameter $\Gamma_{T}$ is defined as
(17)$$\Gamma_{T}=\sum_{k=1}^{N}\int_{t_{k-1}}^{t_{k}}\left[Q_{11}^{\left(k\right)}(Z)\alpha_{S}^{\left(k\right)}(Z)+Q_{12}^{\left(k\right)}(Z)\alpha_{\theta }^{\left(k\right)}(Z)\right]\Delta Tdz=P\times T$$
in which
(18)$$P=\sum_{k=1}^{N}\int_{t_{k-1}}^{t_{k}}\left[Q_{11}^{\left(k\right)}(Z)\alpha_{S}^{\left(k\right)}(Z)+Q_{12}^{\left(k\right)}(Z)\alpha_{\theta }^{\left(k\right)}(Z)\right]dz$$

Using the relationships (5), (6) (10), (12) and (13), the governing Equation (14) is transformed into the following form:(19)$$\begin{array}{l} L_{11}(\psi )+L_{12}(w)+L_{13}(\varphi_{1})+L_{14}(\varphi_{2})=0\\ L_{21}(\psi )+L_{22}(w)+L_{23}(\varphi_{1})+L_{24}(\varphi_{2})=0\\ L_{31}(\psi )+L_{32}(w)+L_{33}(\varphi_{1})+L_{34}(\varphi_{2})=0\\ L_{41}(\psi )+L_{42}(w)+L_{43}(\varphi_{1})+L_{44}(\varphi_{2})=0 \end{array}$$
where $L_{ij}(i,j=1,2,\ldots ,4)$ is a differential operator, and is given in Appendix A. 

The set within Equation (19) is the set of basic equations of laminated conical shells with CNT-patterned layers based on STs.

## 4. Solution Procedure

The two end edges of the laminated truncated conical shell are assumed to be simply supported, and to be restrained against expansion longitudinally, while temperature is increased steadily, so that the boundary conditions are $\zeta =-\zeta_{0}$ and ζ=0 [26,48,49]:(20)$$v=w=0,\;\;\;\;\;\varphi_{2}=0,\;\;\;\;M_{S}=0$$

Here, the following denotations are introduced for convenience: $\zeta =\ln\frac{S}{a_{2}}\;$ and $\zeta_{0}=\ln\frac{a_{1}}{a_{2}}$.

The solution for (19) is defined as [47]:(21)$$\begin{array}{l} \psi =C_{1}a_{2}\;e^{(p+1)\zeta }\;\sin\;(\beta_{m}\zeta )\;\cos(\beta_{n}\theta_{1}),\;\;\;\;w=C_{2}e^{p\zeta }\sin\;(\beta_{m}\zeta )\;\cos(\beta_{n}\theta_{1})\\ \varphi_{1}=C_{3}e^{p\zeta }\cos\;(\beta_{m}\zeta )\;\cos(\beta_{n}\theta_{1}),\;\;\;\;\;\varphi_{2}=C_{4}e^{p\zeta }\sin\;(\beta_{m}\zeta )\;\sin(\beta_{n}\theta_{1}) \end{array}$$$\beta_{m}=\frac{m\pi }{\zeta_{0}}$ and $\beta_{n}=\frac{n}{\sin\gamma }$, wherein (m, n) is the buckling temperature mode, p is the unknown parameter that is defined based on the minimum condition of the buckling temperature, and $C_{i}(i=1,2,\ldots ,4)$ represents unknown coefficients. 

By substituting approximation Equation (21) into the set within Equation (19), and then applying the Galerkin method to the resulting equations, one obtains:(22)$$l_{41}u_{1}-\Gamma_{T}l_{T}u_{2}+l_{43}u_{3}+l_{44}u_{4}=0$$
where
(23)$$u_{1}=-\left|\begin{array}{l} l_{12}\;l_{13}\;l_{14}\\ l_{22}\;l_{23}\;l_{24}\\ l_{32}\;l_{33}\;l_{34} \end{array}\right|,\;\;u_{2}=\left|\begin{array}{l} l_{11}\;l_{13}\;l_{14}\\ l_{21}\;l_{23}\;l_{24}\\ l_{31}\;l_{33}\;l_{34} \end{array}\right|,\;\;u_{3}=-\left|\begin{array}{l} l_{11}\;l_{12}\;l_{14}\\ l_{21}\;l_{22}\;l_{24}\\ l_{31}\;l_{32}\;l_{34} \end{array}\right|,\;u_{4}=\left|\begin{array}{l} l_{11}\;l_{12}\;l_{13}\\ l_{21}\;l_{22}\;l_{23}\\ l_{31}\;l_{32}\;l_{33} \end{array}\right|$$
in which $l_{\mathrm{ij}}(i,j=1,2,\ldots ,4)$ and $l_{T}$ are given in Appendix B.

From Equations (17) and (22), the following expression is found for the uniform buckling temperature of laminated truncated conical shells composed of CNT originating layers:(24)$$T_{UBT}^{ST}=\frac{l_{41}u_{1}+\;l_{43}u_{3}+l_{44}u_{4}}{Pu_{2}l_{T}}$$

Considering the problem within the framework of the CT (that is, considering only the relationships in (5)–the governing equations of laminated conical shells with CNT originating layers) one obtains:(25)$$\begin{array}{l} {\bar{L}}_{11}(\psi )+{\bar{L}}_{12}(w)=0\\ {\bar{L}}_{21}(\psi )+{\bar{L}}_{22}(w)=0 \end{array}$$
where
(26)$$\begin{array}{l} {\bar{L}}_{11}(\psi )=e^{-4\zeta }\left(\begin{array}{l} \delta_{1}\frac{\partial^{4}}{\partial \zeta^{4}}+\delta_{2}\frac{\partial^{3}}{\partial \zeta^{3}}+\delta_{3}\frac{\partial^{2}}{\partial \zeta^{2}}+\delta_{4}\frac{\partial }{\partial \zeta }+\delta_{5}\frac{\partial^{4}}{\partial \theta_{1}^{4}}+\delta_{6}\frac{\partial^{4}}{\partial \zeta^{2}\partial \theta_{1}^{2}}+\delta_{7}\frac{\partial^{3}}{\partial \zeta \partial \theta_{1}^{2}}\\ +\delta_{8}\frac{\partial^{2}}{\partial \theta_{1}^{2}}+\left(\frac{\partial^{2}}{\partial \zeta^{2}}-\frac{\partial }{\partial \zeta }\right)a_{2}e^{\zeta }\cot\gamma \end{array}\right)\\ {\bar{L}}_{12}(w)=e^{-4\zeta }\left[\begin{array}{l} -\delta_{9}\frac{\partial^{4}}{\partial \theta_{1}^{4}}-\delta_{10}\frac{\partial^{4}}{\partial \zeta^{2}\partial \theta_{1}^{2}}+\delta_{11}\frac{\partial^{3}}{\partial \zeta \partial \theta_{1}^{2}}-\delta_{12}\frac{\partial^{2}}{\partial \theta_{1}^{2}}-\delta_{13}\frac{\partial^{4}}{\partial \zeta^{4}}+\delta_{14}\frac{\partial^{3}}{\partial \zeta^{3}}\\ +\delta_{15}\frac{\partial^{2}}{\partial \zeta^{2}}+\delta_{16}\frac{\partial }{\partial \zeta }-Pa_{2}^{2}e^{2\zeta }\left(\frac{\partial^{2}}{\partial \zeta^{2}}-\frac{\partial }{\partial \zeta }\right)T \end{array}\right]\\ {\bar{L}}_{21}(\psi )=e^{-4\zeta }\left(\begin{array}{l} \Delta_{1}\frac{\partial^{4}}{\partial \theta_{1}^{4}}+\Delta_{2}\frac{\partial^{4}}{\partial \zeta^{2}\partial \theta_{1}^{2}}-\Delta_{3}\frac{\partial^{3}}{\partial \zeta \partial \theta_{1}^{2}}+\Delta_{4}\frac{\partial^{2}}{\partial \theta_{1}^{2}}+\Delta_{5}\frac{\partial^{4}}{\partial \zeta^{4}}+\Delta_{6}\frac{\partial^{3}}{\partial \zeta^{3}}+\Delta_{7}\frac{\partial^{2}}{\partial \zeta^{2}}\\ +\Delta_{8}\frac{\partial }{\partial \zeta }-\Delta_{9}\frac{\partial^{4}}{\partial \theta_{1}^{4}} \end{array}\right)\\ {\bar{L}}_{22}(w)=e^{-4\zeta }\left[\begin{array}{l} \Delta_{10}\frac{\partial^{4}}{\partial \zeta^{2}\partial \theta_{1}^{2}}+\Delta_{11}\frac{\partial^{3}}{\partial \zeta \partial \theta_{1}^{2}}+\Delta_{12}\frac{\partial^{2}}{\partial \theta_{1}^{2}}-\Delta_{13}\frac{\partial^{4}}{\partial \zeta^{4}}+\Delta_{14}\frac{\partial^{3}}{\partial \zeta^{3}}+\Delta_{15}\frac{\partial^{2}}{\partial \zeta^{2}}\\ +\Delta_{16}\frac{\partial }{\partial \zeta }+a_{2}e^{\zeta }\cot\gamma \left(\frac{\partial^{2}}{\partial \zeta^{2}}-\frac{\partial }{\partial \zeta }\right) \end{array}\right] \end{array}$$

Similarly, substituting the first two approximation functions from (21) into (25), and then applying the Galerkin method to the resulting equations, the following expression for the uniform buckling temperature for CNT shaped laminated conical shells based on the CT is obtained [41]:(27)$$T_{UBT}^{CT}=\frac{{\bar{l}}_{1}\times {\bar{l}}_{4}+{\bar{l}}_{2}\times {\bar{l}}_{3}}{{\bar{l}}_{3}\times {\bar{l}}_{T}\times P}$$
where ${\bar{l}}_{j}(j=1,2,\ldots ,)$ is the parameter depending on the CNT-shaped laminated conical shell characteristics based on the CT, and ${\bar{l}}_{T}=l_{T}a_{2}^{4}$ is the thermal parameter (both are presented in Appendix C).

## 5. Results and Discussion

### 5.1. Comparative Studies

To check the accuracy of the expressions obtained for the uniform buckling temperature, a comparison is made with the results of the single-layer homogeneous isotropic truncated conical shell, which is presented in Ref. [50] (see Table 1). The data used in the comparison are taken from Ref. [50], and are as follows: $Y_{m}^{\left(1\right)}=7\times 10^{10}\;\mathrm{Pa},\;\;\alpha_{m}^{\left(1\right)}=23\times 10^{-6}(1/K),\;\;\nu_{m}^{\left(1\right)}=0.3$, $\gamma =30^{\circ }$. To compare the results of Ref. [50], the expression (28) was multiplied by $\alpha_{m}^{\left(1\right)}\times 10^{3}$. In addition, $E_{\theta }^{\left(1\right)}=E_{S}^{\left(1\right)}=Y_{m}^{\left(1\right)},\;\;\nu_{S\theta }^{\left(1\right)}=\nu_{\theta S}^{\left(1\right)}=\nu^{\left(1\right)},\;\;\alpha_{\theta }^{\left(1\right)}=\alpha_{S}^{\left(1\right)}=\alpha_{m}^{\left(1\right)}$ are considered in the comparison. It is seen that the magnitudes of UBT ($\alpha_{m}^{\left(1\right)}T_{UBT}^{CT}\times 10^{3}$) are in good agreement with the results of Ref. [50].

### 5.2. Thermal Buckling Analysis

In this subsection, thermal buckling analyses are presented for laminated conical shells consisting of CNT originating layers under uniform temperature rise. The properties of the nanocomposite composed of CNT-reinforced polymethyl methacrylate (PMMA) are given in Table 2 (see, Shen [51]).

The transverse shear stress functions are defined as: Par-TSS functions, or ${\bar{f}}_{i}^{\left(k\right)}(z)=1-4Z^{2}(i=1,2)$, or Cos-Hyp-TSS functions, or ${\bar{f}}_{i}^{\left(k\right)}(z)=\cosh\left(Z\right)-\cosh(1/2)$, or U-TSS functions, or ${\bar{f}}_{i}^{\left(k\right)}(z)=1$ [5,51]. The following definition applies here: ${\bar{f}}_{i}^{\left(k\right)}(z)=\frac{df_{i}^{\left(k\right)}(z)}{dz}$. The uniform buckling temperatures of laminated nanocomposite truncated conical shells within ST and CT are found by minimizing Equations (24) and (27) versus *m*, *n*, and *p*. The lowest values of buckling temperature for laminated nanocomposite cones within ST and CT are achieved at approximately *p* = 2.1, and the number of longitudinal waves is equal to one for all cases. The cross-section types of laminated conical shells, as well as the cross section of the (0°)-single-layer conical shells patterned by CNTs, which are used in the comparison, are shown in Figure 3. In this subsection, the percentages are obtained from the following expressions:(28)$$\frac{T_{UBT}^{CT}-T_{UBT}^{ST}}{T_{UBT}^{CT}}\times 100\%,\;\;\frac{T_{UBT}^{FG}-T_{UBT}^{U}}{T_{UBT}^{U}}\times 100\%,\;\;\;\frac{T_{UBT}^{Lam}-T_{UBT}^{Monolay}}{T_{UBT}^{Monolay}}\times 100\%$$

The variations of the magnitudes of uniform buckling temperature or UBT for (0°) single-layer and laminated truncated conical shells composed of U-, V-, O- and X –originating layers at various transverse shear stress functions, such as Par-TSS, Cos-Hyp-TSS, U-TSS functions, and within CT versus the half-peak angle γ, are tabulated in Table 3 and Table 4. The following data are used in the analysis: $L/r_{1}=0.5,\;r_{1}/h=25,\;h=0.002m,\;V_{cn}^{*(k)}=0.12,\;\;k=1,3,4,p=2.1$. The magnitudes of the UBT slightly decrease in the framework of both theories, while the corresponding numbers of circumferential wave vary depending on the number and arrangement of the layers as γ increments. In the framework of the above three TSS functions, when UBT values for all arrays of laminated conical shells are compared (although almost the same results are obtained for the Par-and Cos-Hyp-TSS functions) the magnitudes of UBT for U-TSS function are different for some layer arrays. The difference between the UBT values at the Par- and U-TSS functions is more pronounced, especially in the laminated conical shells starting with the (0°)-array. For example: the differences between the values of UBT for (0°/90°/0°)-array cones, consisting of U-, V-, O- and X-shaped layers within two theories, are 6.34%, 4.46%, 4.79%, and 8% when $\gamma =10^{\circ }$. However, those differences are 7.3%, 5.09%, 5.17%, and 9.98%, respectively, when $\gamma =30^{\circ }$. It should be emphasized that when the (0°/90°/90°/0°)-sequence cones are compared with the (0°/90°/0°)-sequence cones, the difference is significant. When the Par- and U-TSS functions are used, the least difference between the UBT values occurs in the (90°/0°/90°)-array conical shell, followed by the (90°/0°/0°/90°)-array conical shell. 

While the effect of shear deformations on the UBT values in three- and four-layer shells decreases significantly compared to single-layer shells, it is more pronounced in shells starting with the array starting 90°. When laminated conical shells are compared among themselves, the greatest transverse shear stress effects on UBT values occur in (0°/90°/0°) -array shells with U-, V-, O- and X-patterns, and represent values of 40.72%, 23.19%, 23.85% and 52.3% when $\gamma =10^{\circ }$. Those effects slightly increase, and are 41.62%, 24.65%, 24.66%, and 53.54% when $\gamma =30^{\circ }$. In (90°/0°/90°)-array conical shells, the TSS effects are the lowest, and are 3.77%, 7.61%, 6.01%, and 5.64% when $\gamma =10^{\circ }$. However, those effects are 3.52%, 7.98%, 5.75%, and 4.89% when $\gamma =30^{\circ }$.

The largest and lowest pattern effects on UBT values occur in the (90°/0°/90°)-array cones consisting of X-shaped layers, while the greatest effect is 42.88% when $\gamma =10^{\circ }$. The lowest effect is determined when an amount of (−0.25%) is obtained in the (0°/90°/0°/90°)-array cones when $\gamma =30^{\circ }$ (within ST). In the O-pattern, the greatest effect (−30.43%) is observed in the (90°/0°/90°)-shaped cones, while the lowest effect is observed in (0°/90°/0°)-shaped cones (−11.92%) when $\gamma =10^{\circ }$ (within ST).

When we compare all laminated and (0°)-single-layer conical shells for the Par-TSS (or Cos-Hyp-TSS) function, the biggest differences between UBT values are found in the U-, V-, and O-shaped (0°/90°/0°/90°)-array cone when $\gamma =10^{\circ }$ (which are the values of 47.66%, 34.29%, and 88.71%). While in the X-pattern, it occurs when the (90°/0°/90°)-array cone obtains a value of (−23.7%) at $\gamma =30^{\circ }$ (Table 3). When comparing all laminated and (0°)-single-layer conical shells for the U-TSS function, the biggest differences between UBT values are found in the U-, V-, and O-shaped (0°/90°/0°/90°)-array conical shell when $\gamma =10^{\circ }$ (which are the values of 44.93%, 30.81%, and 83.36%). While in the X-pattern, it occurs when the (90°/0°/90°)-array conical shell obtains a value of (−30.84%) at $\gamma =30^{\circ }$ (Table 4). As can be seen from Table 3 and Table 4, the influence of the arrangement and number of layers on the buckling temperature is reduced when using U-TSS compared to Par-TSS (or Cos-Hyp-TSS) functions in U-, V- and O-shaped conical shells. This effect is more pronounced in nanocomposite conical shells with an X-shaped pattern.

The variations of the magnitudes of UBT for (0°)-single-layer and laminated truncated conical shells composed of U-, V-, O- and X –originating layers within ST and CT (versus $r_{1}/h$) are tabulated in Table 5 and Figure 4, Figure 5 and Figure 6. The laminated truncated conical shells have different layer sequences and consist of three and four layers. The following data are valid in the analysis: $L/r_{1}=0.5,\;\gamma =15^{\circ },\;h=0.002\;m,\;V_{cn}^{*(k)}=0.28,\;k=1,3,4,\;p=2.1$. The transverse shear stress function is considered as the cosine-hyperbolic function. The magnitudes of UBT decrease in the framework of ST for all TSSs (within the CT), while the corresponding numbers of circumferential wave vary depending on the number and arrangement of the layers as $r_{1}/h$ increments. When the $r_{1}/h$ ratio increases, the effects of the heterogeneity on the buckling temperature of laminated cones consisting of V-, O-, X-shaped layers are changed significantly compared to U-pattern laminated cones, and those effects differ according to the arrangement and number of the layers. For example, in the Cos-Hyp-TSS function, the values of the effects of V-pattern on the UBT for (0°), (0°/90°/0°), (90°/0°/90°), (0°/90°/0°/90°), (0°/90°/90°/0°), and (90°/0°/0°/90°)-array cones, compared to U-shaped cones, are (−10.42%), (−17.63%), (−3.91%), (−13.32%), (−23.17%), and (−4.79%), respectively (when $r_{1}/h=20$). However, the values of those effects are (−18.46%), (−24.89%), (−10.98%), (−12.99%), (−29.25%), and (−9.18%), respectively, when $r_{1}/h=35$. The effects of the X-model on the UBT of laminated cones with array and numbered layers discussed above are (+5.16%), (−16.21%), (+61.8%), (−3.52%), (−19.66%), and (+48.4%), respectively, when $r_{1}/h=20$. However, these are (+20.77%), (−4.27%), (+76.37%), (+0.82%), (−7.77%), and (+62.56%), respectively, when $r_{1}/h=35$. The most significant value of the effect on UBT in cones with O-shaped layers is (−26.34%) for a (90°/0°/90°)-array cone when $r_{1}/h=20$. That effect increases to a value of (−34.61%) when $r_{1}/h=35$. On the other hand, in laminated shells with O-shaped layers, the lowest value for pattern effect on the UBT for a (0°/90°/0°)-array is (+3.843%) when $r_{1}/h=20$. However, it is a value of (−8.252%), and occurs in the (0°/90°/90°/0°)-sequence shell, when $r_{1}/h=35$. 

A significant decrease is observed in the effect of shear stresses on the UBT when the $r_{1}/h$ ratio increases. Among the five laminated shells, the highest shear stress effect on the UBT–a value of 70.87%–occurs in the (0°/90°/0°)-array shell, with the X-shaped layers, when $r_{1}/h=20$. However, the weakest value of effect is 2.14%, which occurs in the (90°/0°/90°)-sequence shell (Figure 4). Also, the use of laminated conical shells reduces the effects of shear stresses on the UBT compared to (0°)-single-layer shells. Concerning the laminated cones starting with the array starting 0°, the effect of shear stresses on the buckling temperature shows a significant decrease in laminated cones starting with the array starting 90°. For example: while the value of effect of shear stresses on the magnitudes of UBT is 33.84% in the (0°/90°/0°/90°)-array shells consisting of V-shaped layers, it is a value of 15.57% in the (90°/0°/0°/90°)-array shells (Figure 5). Depending on the increase in $r_{1}/h$ ratio, the effect of arrangement and number of layers on the UBT shows significant changes compared to the (0)-single-layer shell. The most change occurs in the UBT of the (0°/90°/0°/90°)-array conical shell consisting of X-shaped layers when compared to the (0°)-single-layer conical shell. For instance, the difference between buckling temperatures is 22.31% when $r_{1}/h=20$, while this difference is (−2.39%) when $r_{1}/h=35$ (Figure 6).

## 6. Conclusions

The thermal buckling of laminated truncated conical shells composed of CNT originating layers within STs is studied. The modified Donnell type shell theory is applied to derive the basic equations, and then the Galerkin method is applied to the basic equations to find a new expression for the UBT of laminated truncated conical shells composed of CNT originating layers (within ST and CT). Four types of single-walled carbon nanotube distributions across the thickness of the layers are considered (namely uniform and functionally graded). The Par-, Cos-Hyp- and U-transverse shear stress functions are used in the analysis. The influences of change in CNT models, and the arrangement and number of the layers on the UBT using different shear stress functions, are examined.

Numerical analyses revealed the following generalizations:

(a)The magnitudes of the UBT slightly decrease in the framework of both theories, while the corresponding numbers of circumferential waves vary depending on the number and arrangement of the layers as γ increments.(b)When the UBT values for all arrays of laminated conical shells are compared, although almost the same results are obtained for the Par-and Cos-Hyp-TSS functions, the magnitudes of UBT for U-TSS function are different for some layer arrays. (c)The difference between the UBT values at Par- and U-TSS functions is more pronounced, especially in the laminated conical shells starting with the (0°)-array. (d)When the Par- and U-TSS functions are used, the least difference between the UBT values occurs in the (90°/0°/90°)-array conical shell, followed by the (90°/0°/0°/90°)-array conical shell.(e)While the effect of shear deformations on the UBT values in three- and four-layer shells decreases significantly compared to single-layer shells, it is more pronounced in shells starting with the array starting 90°. (f)The largest and lowest pattern effects on UBT values occur in the (90°/0°/90°)-array cones consisting of X-shaped layers.(g)When the $r_{1}/h$ ratio increases, the effects of the heterogeneity on the buckling temperature of cones consisting of V-, O-, and X-shaped layers are changed significantly compared to U-pattern laminated cones, and those effects differ according to the arrangement and number of the layers. (h)A significant decrease is observed in the effect of shear stresses on the UBT when the $r_{1}/h$ ratio increases. (i)Among the five laminated shells, the highest shear stress effect on the UBT occurs in the (0°/90°/0°)-array shell with the X-shaped layers, while the weakest effect occurs in the (90°/0°/90°)-sequence shell. (j)The use of laminated conical shells reduces the effects of shear stresses on the UBT compared to (0°)-single-layer shells. (k)Depending on the increase of the $r_{1}/h$ ratio, the effect of arrangement and number of layers on the UBT shows significant changes compared to the (0)-single-layer shell. 

Since laminated heterogeneous nanocomposite conical shells, reinforced with carbon nanotubes with robust heat resistance and high strength, are frequently used in modern aerospace and rocket technology, shipbuilding, energy and chemical engineering, and other fields exposed to very high temperatures, the results obtained in this research on their thermal buckling behavior should be considered during design.

## Figures and Tables

**Figure 1 materials-15-07427-f001:**
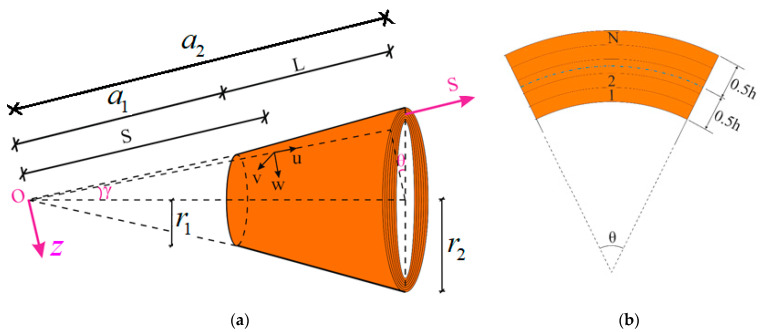
(**a**) Laminated truncated conical shell composed of CNT originating layers; (**b**) the cross section.

**Figure 2 materials-15-07427-f002:**
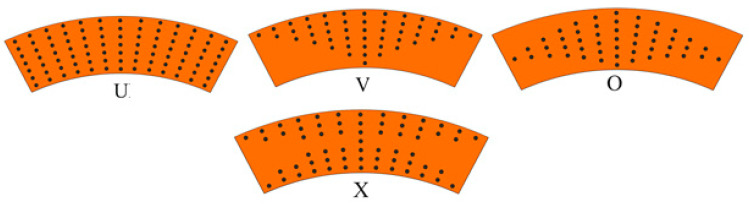
Distribution of volume fractions of CNTs across the thickness of the layer.

**Figure 3 materials-15-07427-f003:**
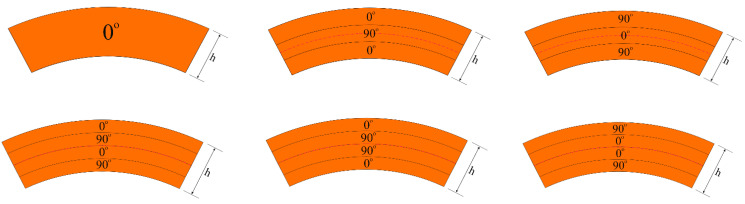
The cross-section types of laminated and single-layer truncated conical shell.

**Figure 4 materials-15-07427-f004:**
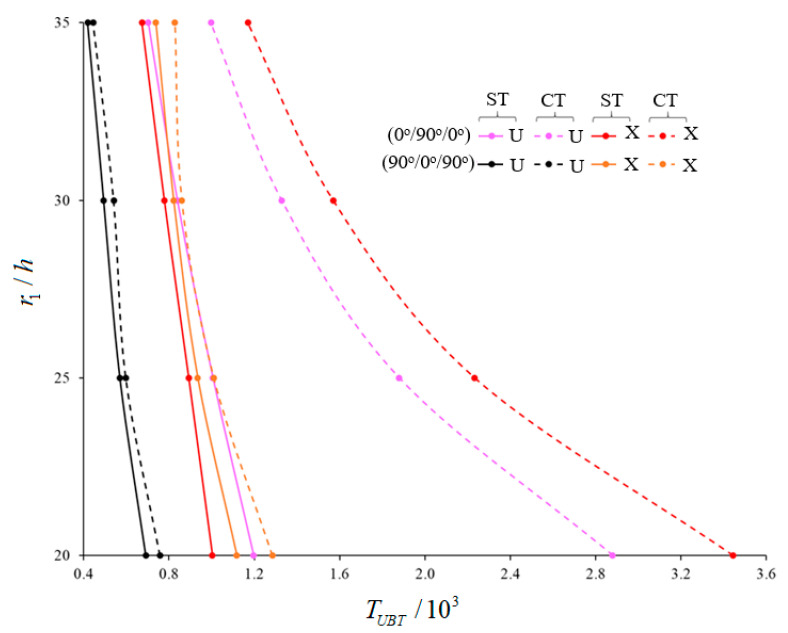
Variations of UBT for (90°/0°/90°)- and (0°/90°/0°)-array laminated cones with U- and X- shaped layers for Cos-Hyp-TSS function (within CT versus $r_{1}/h$.

**Figure 5 materials-15-07427-f005:**
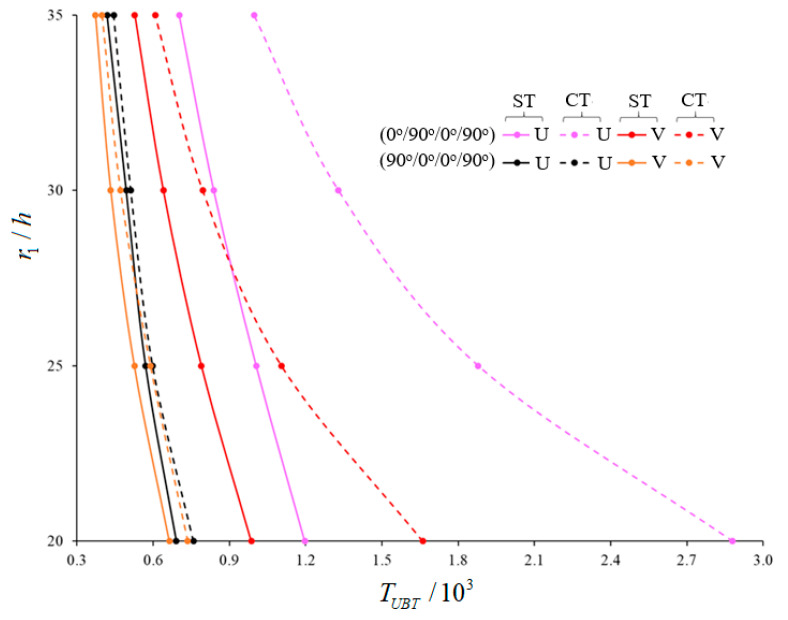
Variations of UBT for (0°/90°/0°/90°)- and (90°/0°/0°/90°)-array laminated cones with U- and V-shaped layers for Cos-Hyp-TSS function (within CT versus $r_{1}/h$.

**Figure 6 materials-15-07427-f006:**
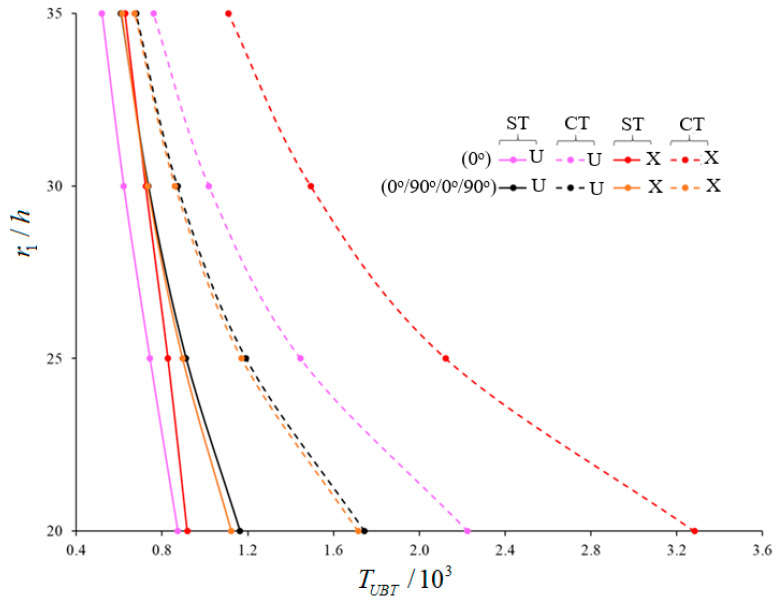
Variations of UBT for (0°)-single-layer and (0°/90°/0°/90°)-array laminated cones with U- and X -shaped layers for Cos-Hyp-TSS function (within CT versus $r_{1}/h$.

**Table 1 materials-15-07427-t001:** Comparison of magnitudes of $T_{UBT}^{CT}$ with the results of Ref. [50].

	$\alpha_{m}^{\left(1\right)}T_{UBT}^{CT}\times 10^{3},\;(m,n\;)$				
$\left(r_{1}+r_{2}\right)/2h$	200	300	400	500	600
Ref. [50]	2.25	1.50	1.13	0.90	0.75
Current study	2.246 (7, 12)	1.493 (26,4)	1.12 (27, 11)	0.895 (28, 16)	0.746 (29,19)

**Table 2 materials-15-07427-t002:** Properties of nanocomposites and efficiency parameters in the layers.

Thermo-Mechanical Properties of CNTin the Layer	Thermo-Mechanical Properties of PMMA in the Layer	The Efficiency Parameters in the Layer
$\begin{array}{l} Y_{Scn}^{\left(k\right)}=(6.3998-4.338417\times 10^{-3}\times T\\ +7.43\times 10^{-6}\times T^{2}-4.458333\times 10^{-9}\times T^{3})\times 10^{12}\\ Y_{\theta cn}^{\left(k\right)}=(8.02155-5.420375\times 10^{-3}\times T\\ +9.275\times 10^{-6}\times T^{2}-5.5625\times 10^{-9}\times T^{3})\times 10^{12}\\ G_{S\theta cn}^{\left(k\right)}=(1.4075+3.476208\times 10^{-3}\times T\\ -6.965\times 10^{-6}\times T^{2}+4.479167\times 10^{-9}\times T^{3})\times 10^{12}\\ \alpha_{Scn}^{\left(k\right)}=(-1.12515+0.02291688\times T\\ -2.887\times 10^{-5}\times T^{2}+1.13625\times 10^{-8}\times T^{3})\times 10^{-6}\\ \alpha_{\theta cn}^{\left(k\right)}=(5.43715-0.98462510^{-4}\times T\\ +2.9\times 10^{-7}\times T^{2}+1.25\times 10^{-11}\times T^{3})\times 10^{-6} \end{array}$	$Y_{m}^{\left(k\right)}=(3.52-0.0034T)\;\mathrm{Gpa},$ $\nu_{m}^{\left(k\right)}=0.34$ $\alpha_{m}^{\left(k\right)}=45(1+0.0005\Delta T)\times 10^{-6}/K$ at *T* = 300 K $\;Y_{m}^{\left(k\right)}=2.5\;\mathrm{Gpa}$, $\alpha_{m}^{\left(k\right)}=45\times 10^{-6}/K$ (k=1,2,…,N)	$\begin{array}{l} \eta_{1}^{\left(k\right)}=0.137,\;\;\\ \eta_{2}^{\left(k\right)}=1.022,\;\\ \eta_{3}^{\left(k\right)}=0.7\eta_{2}^{\left(k\right)}\;\;\;\;\\ at\;\;\;V_{cn}^{*(k)}=0.12;\;\\ \eta_{1}^{\left(k\right)}=0.142,\;\\ \eta_{2}^{\left(k\right)}=1.626,\;\\ \eta_{3}^{\left(k\right)}=0.7\eta_{2}^{\left(k\right)}\\ at\;\;\;V_{cn}^{*(k)}=0.17;\\ \eta_{1}^{\left(k\right)}=0.141,\;\;\\ \eta_{2}^{\left(k\right)}=1.585,\\ \eta_{3}^{\left(k\right)}=0.7\eta_{2}^{\left(k\right)}\\ at\;\;\;\;V_{cn}^{*(k)}=0.28 \end{array}$
The geometrical properties of CNT		
$a_{cn}=9.26\;\mathrm{nm},\;r_{cn}=0.68\;\mathrm{nm},\;h_{cn}=0.067\;\mathrm{nm}$, $\nu_{S\theta }^{cn}=0.175$		

**Table 3 materials-15-07427-t003:** Variations of UBT for single-layer and laminated cones with CNT-shaped layers for Par- and Cos-Hyp-SS functions (within CT versus γ).

		$T_{UBT}/10^{3}\;(n_{cr})\mathrm{for}\;\mathrm{Par}-\;\mathrm{and}\;\mathrm{Cos}-\mathrm{Hyp}-\mathrm{TSS}\;\mathrm{Functions}$							
γ	Arrangement of Layers	U		V		O		X	
		ST	CT	ST	CT	ST	CT	ST	CT
10°	(0°)	0.789 (5)	1.375 (4)	0.662 (4)	0.972 (4)	0.549 (5)	0.754 (4)	0.932 (5)	1.991 (4)
	(0°/90°/0°)	1.015 (5)	1.712 (5)	0.815 (5)	1.061 (5)	0.894 (5)	1.174 (5)	0.977 (5)	2.048 (5)
	(90°/0°/90°)	0.562 (3)	0.584 (3)	0.534 (4)	0.578 (4)	0.391 (4)	0.416 (4)	0.803 (3)	0.851 (3)
	(0°/90°/0°/90°)	0.880 (4)	1.085 (4)	0.802 (4)	0.976 (4)	0.655 (4)	0.758 (4)	0.876 (4)	1.053 (4)
	(0°/90°/90°/0°)	1.165 (5)	1.837 (4)	0.889 (4)	1.038 (4)	1.036 (4)	1.258 (4)	1.082 (5)	2.151 (4)
	(90°/0°/0°/90°)	0.599 (3)	0.645 (3)	0.567 (4)	0.619 (4)	0.429 (4)	0.475 (4)	0.828 (3)	0.895 (3)
20°	(0°)	0.764 (6)	1.348 (4)	0.643 (5)	0.950 (4)	0.531 (5)	0.735 (4)	0.900 (6)	1.956 (4)
	(0°/90°/0°)	0.984 (5)	1.675 (5)	0.780 (5)	1.029 (5)	0.860 (5)	1.138 (5)	0.942 (6)	2.010 (5)
	(90°/0°/90°)	0.519 (3)	0.539 (3)	0.488 (4)	0.529 (4)	0.361 (4)	0.384 (4)	0.740 (3)	0.780 (3)
	(0°/90°/0°/90°)	0.841 (4)	1.042 (4)	0.769 (4)	0.942 (4)	0.624 (4)	0.724 (4)	0.839 (4)	1.010 (4)
	(0°/90°/90°/0°)	1.123 (5)	1.794 (4)	0.849 (5)	1.000 (4)	0.996 (4)	1.213 (4)	1.047 (5)	2.109 (4)
	(90°/0°/0°/90°)	0.563 (3)	0.606 (3)	0.528 (4)	0.578 (4)	0.404 (4)	0.448 (4)	0.773 (3)	0.833 (3)
30°	(0°)	0.739 (7)	1.314 (4)	0.624 (5)	0.921 (3)	0.514 (6)	0.711 (4)	0.866 (7)	1.911 (4)
	(0°/90°/0°)	0.951 (6)	1.629 (5)	0.746 (5)	0.990 (5)	0.825 (5)	1.095 (5)	0.911 (6)	1.961 (5)
	(90°/0°/90°)	0.466 (3)	0.483 (3)	0.438 (4)	0.476 (4)	0.328 (4)	0.348 (4)	0.661 (3)	0.695 (3)
	(0°/90°/0°/90°)	0.801 (4)	0.992 (4)	0.732 (4)	0.898 (4)	0.589 (4)	0.684 (4)	0.799 (4)	0.961 (4)
	(0°/90°/90°/0°)	1.085 (5)	1.751 (4)	0.803 (5)	0.952 (4)	0.949 (4)	1.156 (4)	1.017 (5)	2.053 (4)
	(90°/0°/0°/90°)	0.517 (3)	0.555 (3)	0.483 (4)	0.532 (4)	0.376 (4)	0.417 (4)	0.705 (3)	0.759 (3)

**Table 4 materials-15-07427-t004:** Variations of UBT for single-layer and laminated cones with CNT-shaped layers for the U- TSS function (within CT versus γ).

		$T_{UBT}/10^{3}\;(n_{cr})\mathrm{for}\;U-\;\mathrm{TSS}\;\mathrm{Function}$							
γ	Arrangement of Layers	U		V		O		X	
		ST	CT	ST	CT	ST	CT	ST	CT
10°	(0°)	0.848 (5)	1.375 (4)	0.701 (4)	0.972 (4)	0.583 (5)	0.754 (4)	1.017 (5)	1.991 (4)
	(0°/90°/0°)	1.084 (5)	1.712 (5)	0.853 (5)	1.061 (5)	0.939 (5)	1.174 (5)	1.062 (5)	2.048 (5)
	(90°/0°/90°)	0.565 (3)	0.584 (3)	0.541 (4)	0.578 (4)	0.396 (4)	0.416 (4)	0.811 (3)	0.851 (3)
	(0°/90°/0°/90°)	0.914 (4)	1.085 (4)	0.834 (4)	0.976 (4)	0.674 (4)	0.758 (4)	0.841 (4)	1.053 (4)
	(0°/90°/90°/0°)	1.229 (4)	1.837 (4)	0.917 (4)	1.038 (4)	1.069 (4)	1.258 (4)	1.166 (5)	2.151 (4)
	(90°/0°/0°/90°)	0.609 (3)	0.645 (3)	0.579 (4)	0.619 (4)	0.440 (4)	0.475 (4)	0.841 (3)	0.895 (3)
20°	(0°)	0.826 (6)	1.348 (4)	0.683 (5)	0.950 (4)	0.565 (5)	0.735 (4)	0.988 (6)	1.956 (4)
	(0°/90°/0°)	1.054 (5)	1.675 (5)	0.820 (5)	1.029 (5)	0.904 (5)	1.138 (5)	1.035 (5)	2.010 (5)
	(90°/0°/90°)	0.523 (3)	0.539 (3)	0.495 (4)	0.529 (4)	0.366 (4)	0.384 (4)	0.746 (3)	0.780 (3)
	(0°/90°/0°/90°)	0.876 (4)	1.042 (4)	0.800 (4)	0.942 (4)	0.643 (4)	0.724 (4)	0.874 (4)	1.010 (4)
	(0°/90°/90°/0°)	1.197 (4)	1.794 (4)	0.880 (4)	1.000 (4)	1.029 (4)	1.213 (4)	1.132 (5)	2.109 (4)
	(90°/0°/0°/90°)	0.573 (3)	0.606 (3)	0.539 (4)	0.578 (4)	0.414 (4)	0.448 (4)	0.785 (3)	0.833 (3)
30°	(0°)	0.803 (6)	1.313 (4)	0.665 (5)	0.921 (3)	0.548 (5)	0.711 (4)	0.963 (6)	1.911 (4)
	(0°/90°/0°)	1.026 (5)	1.628 (5)	0.786 (5)	0.990 (5)	0.870 (5)	1.095 (5)	1.012 (5)	1.962 (5)
	(90°/0°/90°)	0.470 (3)	0.483 (3)	0.443 (4)	0.476 (4)	0.332 (4)	0.348 (4)	0.666 (3)	0.695 (3)
	(0°/90°/0°/90°)	0.836 (4)	0.992 (4)	0.763 (4)	0.898 (4)	0.608 (4)	0.684 (4)	0.834 (4)	0.961 (4)
	(0°/90°/90°/0°)	1.152 (5)	1.737 (4)	0.835 (5)	0.952 (4)	0.983 (4)	1.156 (4)	1.103 (4)	2.053 (4)
	(90°/0°/0°/90°)	0.527 (3)	0.555 (3)	0.494 (4)	0.532 (4)	0.386 (4)	0.417 (4)	0.717 (3)	0.759 (3)

**Table 5 materials-15-07427-t005:** Variations of UBT for single-layer and laminated cones with CNT-shaped layers for the Cos-Hyp-TSS function (within CT versus $r_{1}/h$).

		$T_{UBT}/10^{3}\;(n_{cr})\mathrm{for}\;\mathrm{Cos}-\mathrm{Hyp}-\mathrm{TSS}\;\mathrm{Function}$							
		U		V		O		X	
$r_{1}/h$	Arrangement of Layers	ST	CT	ST	CT	ST	CT	ST	CT
20	(0°)	0.873 (6)	2.225 (4)	0.782 (5)	1.531 (3)	0.700 (5)	1.173 (4)	0.918 (6)	3.283 (3)
	(0°/90°/0°)	1.197 (5)	2.879 (4)	0.986 (5)	1.661 (4)	1.243 (5)	1.938 (4)	1.003 (5)	3.443 (4)
	(90°/0°/90°)	0.691 (3)	0.758 (3)	0.664 (3)	0.736 (3)	0.509 (3)	0.540 (3)	1.118 (3)	1.285 (3)
	(0°/90°/0°/90°)	1.164 (4)	1.744 (3)	1.009 (4)	1.525 (4)	0.914 (4)	1.182 (4)	1.123 (4)	1.715 (3)
	(0°/90°/90°/0°)	1.450 (4)	3.167 (4)	1.114 (4)	1.607 (4)	1.537 (4)	2.159 (4)	1.165 (5)	3.674 (4)
	(90°/0°/0°/90°)	0.752 (3)	0.887 (3)	0.716 (4)	0.848 (3)	0.576 (4)	0.675 (3)	1.116 (3)	1.348 (3)
25	(0°)	0.743 (6)	1.445 (4)	0.635 (5)	1.004 (4)	0.546 (5)	0.772 (4)	0.828 (6)	2.123 (4)
	(0°/90°/0°)	1.006 (5)	1.879 (5)	0.790 (5)	1.105 (5)	0.961 (5)	1.292 (5)	0.892 (6)	2.233 (5)
	(90°/0°/90°)	0.570 (3)	0.598 (3)	0.526 (4)	0.589 (4)	0.392 (4)	0.417 (4)	0.935 (3)	1.010 (3)
	(0°/90°/0°/90°)	0.912 (4)	1.192 (4)	0.799 (4)	1.027 (4)	0.693 (4)	0.811 (4)	0.896 (4)	1.171 (4)
	(0°/90°/90°/0°)	1.203 (5)	2.080 (4)	0.884 (4)	1.090 (4)	1.182 (4)	1.465 (4)	1.021 (5)	2.391 (4)
	(90°/0°/0°/90°)	0.600 (3)	0.659 (3)	0.549 (4)	0.622 (4)	0.433 (4)	0.481 (4)	0.906 (3)	1.008 (3)
30	(0°)	0.621 (6)	1.018 (5)	0.516 (5)	0.715 (4)	0.432 (5)	0.550 (5)	0.724 (6)	1.493 (5)
	(0°/90°/0°)	0.839 (5)	1.329 (5)	0.640 (5)	0.795 (5)	0.760 (5)	0.931 (5)	0.778 (6)	1.570 (5)
	(90°/0°/90°)	0.494 (4)	0.512 (3)	0.433 (4)	0.471 (4)	0.320 (4)	0.332 (4)	0.822 (3)	0.860 (3)
	(0°/90°/0°/90°)	0.735 (4)	0.872 (4)	0.648 (5)	0.756 (4)	0.552 (4)	0.610 (4)	0.732 (4)	0.860 (4)
	(0°/90°/90°/0°)	0.990 (5)	1.488 (5)	0.710 (5)	0.806 (5)	0.937 (5)	1.088 (4)	0.882 (5)	1.692 (5)
	(90°/0°/0°/90°)	0.497 (4)	0.535 (3)	0.444 (4)	0.483 (4)	0.345 (4)	0.368 (4)	0.772 (3)	0.824 (3)
35	(0°)	0.520 (6)	0.760 (5)	0.424 (5)	0.538 (5)	0.348 (6)	0.416 (5)	0.628 (6)	1.109 (5)
	(0°/90°/0°)	0.703 (6)	0.997 (5)	0.528 (6)	0.608 (5)	0.618 (5)	0.714 (5)	0.673 (6)	1.170 (5)
	(90°/0°/90°)	0.419 (4)	0.445 (4)	0.373 (4)	0.399 (4)	0.274 (4)	0.280 (4)	0.739 (4)	0.828 (4)
	(0°/90°/0°/90°)	0.608 (4)	0.679 (4)	0.529 (5)	0.586 (5)	0.457 (4)	0.489 (4)	0.613 (4)	0.672 (4)
	(0°/90°/90°/0°)	0.824 (5)	1.119 (5)	0.583 (5)	0.622 (5)	0.756 (5)	0.841 (5)	0.760 (5)	1.262 (5)
	(90°/0°/0°/90°)	0.414 (4)	0.443 (4)	0.376 (4)	0.399 (4)	0.288 (4)	0.300 (4)	0.673 (4)	0.713 (3)

## Data Availability

No data were reported in the study.

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
