# Peer review of "On the Solution of Thermal Buckling Problem of Moderately Thick Laminated Conical Shells Containing Carbon Nanotube Originating Layers"

_materials, 2022, doi:10.3390/ma15217427_

Round 1

Reviewer 1 Report

This paper examined the buckling behavior of the moderately thick laminated conical shells containing CNT originating layers under uniform temperature rise. The buckling temperatures obtained from without shear deformation and three different types of shear deformation theoretical frameworks, i.e., Par-TSS, Cos-Hyp-TSS and U-TSS, are compared. Different distribution forms of CNT along the plate thickness direction are considered, such as U, V, O and X, as well the cross section types is discussed. Moreover, the effects of volume fraction and the dimension of structure on the thermal buckling behavior are also investigated. However, the results gained are limited, and the types and explanations of figures and tables are relatively simple. Furthermore, some obvious errors in the sentences still exist. Thus, the manuscript is not recommended to Materials for publication, and some suggestions are advised for further revision:

(1)  The data in Table 3 include the results obtained by Par-TSS and Cos-Hyp-TSS, but only one set of values exists. Besides that, reference should be provided to demonstrate the validation of the results, and more results should be also added to explain the difference between the theoretical frameworks.

(2)  The effect of volume fraction on the buckling temperature is not clearly studied, and a comparison should be used to discuss the relevant results. Otherwise, statement should be modified in the abstract to keep consistent.

(3)  The results obtained by different theoretical framework should be clearly explained, for example, what is the reason about the difference between [0°/90°/0°] and [90°/0°/90°]. In addition, the obtained conclusion should be revised, involving the scope of application for the three shear deformation theories in the field of engineering.

(4)  The manuscript should be polished by careful editing and paying attention to English spelling, format, sentence structure. For exmples, the beginning should be written as 2.1 of section 2, the origin is “O” or “O”? keyword is “laminated truncated conical shell” or “laminated conical shell”? the explanation of symbols in Eq. (3) omits density ρ, the “O-” or “O-” about the distribution forms of CNT in Eq. (4), where is “j” in Lij (i=1, 2, …, 4), Eqs. (18) or Eq. (18)? Note the location of abbreviations UBT and CT.

Author Response

EXPLANATION TO REVIEWER 1:

First of all, we would like to thank the Reviewer 1 for his/her improving remarks and the time spent for them.

Comments and Suggestions for Authors

This paper examined the buckling behavior of the moderately thick laminated conical shells containing CNT originating layers under uniform temperature rise. The buckling temperatures obtained from without shear deformation and three different types of shear deformation theoretical frameworks, i.e., Par-TSS, Cos-Hyp-TSS and U-TSS, are compared. Different distribution forms of CNT along the plate thickness direction are considered, such as U, V, O and X, as well the cross section types is discussed. Moreover, the effects of volume fraction and the dimension of structure on the thermal buckling behavior are also investigated. However, the results gained are limited, and the types and explanations of figures and tables are relatively simple. Furthermore, some obvious errors in the sentences still exist. Thus, the manuscript is not recommended to Materials for publication, and some suggestions are advised for further revision:

SUGGESTION 1: The data in Table 3 include the results obtained by Par-TSS and Cos-Hyp-TSS, but only one set of values exists. Besides that, reference should be provided to demonstrate the validation of the results, and more results should be also added to explain the difference between the theoretical frameworks.

EXPLANATION 1: Thanks. It is noted that Table 3 and Table 4 are parts of the same table. Since the table is large, we divided it into two parts as Tables 3 and 4. You are right, when Cos-Hyp- and Par-transverse shear stress (Cos-Hyp-TSS and Par-TSS) functions are used as shear stress functions, a value set is given because the critical temperature values ​​differ in the 4th and fifth digits of the comma (Table 1* for Reviewers). Mathematically, this means that since the numerical values ​​of the integrals in terms of these functions are almost the same, the values ​​of the buckling temperature are also the same. Therefore, it turns out that approximately the same results are obtained when Cos-Hyp-TSS and Par-TSS functions are used. This shows that using either of the Cos-Hyp-TSS and Par-TSS functions is more reliable than using the U-TSS function. This is not contrary to the general application principles of shear deformation theory. It should be emphasized that the obtained formulas are derived for the general shear stress functions fi(z), (i=1,2). In numerical analysis, detailed analyzes were made by choosing three different shear stress functions.

Table 1*(for Reviewers) Variations of UBT for single-layer and laminated cones with CNT -shaped layers for Par- and Cos-Hyp-SS functions for  = 10°

= 10°

 (ncr) for Par (m=1)

Arrangement  of  layers

U

V

ST

CT

ST

CT

(0o)

0.7886089971 (5)

1.375054951 (4)

0.6623625174 (4)

0.9721811089 (4)

(0o/90o/0o)

1.015253916 (5)

1.712211543 (5)

0.8150226251 (5)

1.060832094 (5)

(90o/0o/90o)

0.5615988629 (3)

0.5844530616 (3)

0.5342772206 (4)

0.5781959494 (4)

(0o/90o/0o/90o)

0.8795123466 (4)

1.085330656 (4)

0.8024069521 (4)

0.9760816848 (4)

(0o/90o/90o/0o)

1.165394406 (5)

1.837398691 (4)

0.888920668 (4)

1.037713209 (4)

(90o/0o/0o/90o)

0.5991216424 (3)

0.6451522084 (3)

0.5674921594 (4)

0.6188677499 (4)

Arrangement  of  layers

O

X

ST

CT

ST

CT

(0o)

0.5526068956 (4)

0.7537447114 (4)

0.9320270558 (5)

1.991222944 (4)

(0o/90o/0o)

0.8944756978 (5)

1.173700302 (5)

0.9767367035 (5)

2.047765193 (5)

(90o/0o/90o)

0.3908647278 (4)

0.4162628424 (4)

0.8033860213 (3)

0.8511516793 (3)

(0o/90o/0o/90o)

0.6552096035 (4)

0.7580773693 (4)

0.8759402171 (4)

1.052976572 (4)

(0o/90o/90o/0o)

1.036114187 (4)

1.258322092 (4)

1.081848047 (5)

2.151361554 (4)

(90o/0o/0o/90o)

0.4288874994 (4)

0.475280073 (4)

0.8279833725 (3)

0.8949272732 (3)

Cos-ST (m=1)

Arrangement  of  layers

U

V

ST

CT

ST

CT

(0o)

0.7889998522 (5)

1.375054951 (4)

0.6626227572 (4)

0.9721811089 (4)

(0o/90o/0o)

1.015683028 (5)

1.712211543 (5)

0.8152719186 (5)

1.060832094 (5)

(90o/0o/90o)

0.5616339913 (3)

0.5844530616 (3)

0.5343308452 (4)

0.5781959494 (4)

(0o/90o/0o/90o)

0.8797749139 (4)

1.085330656 (4)

0.8026408237 (4)

0.9760816848 (4)

(0o/90o/90o/0o)

1.165765822 (5)

1.837398691 (4)

0.8890898234 (4)

1.037713209 (4)

(90o/0o/0o/90o)

0.5992169554 (3)

0.6451522084 (3)

0.5675930639 (4)

0.6188677499 (4)

Arrangement  of  layers

O

X

ST

CT

ST

CT

(0o)

0.5528578285 (4)

0.7537447114 (4)

0.9325621293 (5)

1.991222944 (4)

(0o/90o/0o)

0.894787302 (5)

1.173700302 (5)

0.9772611732 (5)

2.047765193 (5)

(90o/0o/90o)

0.390910039 (4)

0.4162628424 (4)

0.8034381476 (3)

0.8511516793 (3)

(0o/90o/0o/90o)

0.6553527079 (4)

0.7580773693 (4)

0.8762123912 (4)

1.052976572 (4)

(0o/90o/90o/0o)

1.036318973 (4)

1.258322092 (4)

1.08233884 (5)

2.151361554 (4)

(90o/0o/0o/90o)

0.4289859943 (4)

0.475280073 (4)

0.82809336 (3)

0.8949272732 (3)

Note 1: Other approximation functions are also available in the literature:

(Table 2*for Reviewers)

Note 2: Apart from these models, there are some models.

SUGGESTION 2: The effect of volume fraction on the buckling temperature is not clearly studied, and a comparison should be used to discuss the relevant results. Otherwise, statement should be modified in the abstract to keep consistent.

EXPLANATION 2: Thank you. You are right. The necessary modification is made in the summary part.

SUGGESTION 3: The results obtained by different theoretical framework should be clearly explained, for example, what is the reason about the difference between [0°/90°/0°] and [90°/0°/90°]. In addition, the obtained conclusion should be revised, involving the scope of application for the three shear deformation theories in the field of engineering.

EXPLANATION 3:  Thanks. One of the main goals of this study was to study the influence of arrangement and number of layers on the buckling temperature. For this reason, numerical calculations were made by changing the layer arrangement and number in all tables and their effects were evaluated in detail. [0°/90°/0°] and [90°/0°/90°] or [0°/90°/90°/0°] and [90°/0°/0°/90°/] the reason for the difference in their configurations is that the structure (and especially the rigidity-stiffness) of the structural element changes when the arrangement and number of layers change. A structural element with a different layer arrangement and number is a new element, it cannot be used in place of another. These features can be determined during design, what arrangement and number of structural elements should be used depending on the conditions of use in engineering applications.

SUGGESTION 4: The manuscript should be polished by careful editing and paying attention to English spelling, format, sentence structure. For examples, the beginning should be written as 2.1 of section 2, the origin is “O” or “O”? keyword is “laminated truncated conical shell” or “laminated conical shell”? the explanation of symbols in Eq. (3) omits density ρ, the “O-” or “O-” about the distribution forms of CNT in Eq. (4), where is “j” in Lij (i=1, 2, …, 4), Eqs. (18) or Eq. (18)? Note the location of abbreviations UBT and CT.

EXPLANATION 4: Thanks. You are right. All your suggestions have been taken into account and corrected in the revised manuscript.

The beginning is written as 2.1 of section 2

Instead of origin “O”, origin “O” is written in italic.

Keyword is “laminated truncated conical shell” instead of “laminated conical shell”?

Since the material density is not included in the basic equations in thermal buckling problems, it is not included in the mathematical operations and was not used. Spelling error corrected.

CNT distribution is denoted by “O-” instead of “O-”.

Lij (i, j=1, 2, …, 4) is written instead of Lij (i=1, 2, …, 4).

Since (18) is a system of equations or a set of equations, “set of equations (18)” is more correct. This sentence has been modified as follows

The set of equations (18) is the set of basic equations of laminated conical shells with CNT patterned layers based on STs.

UBT and CT abbreviations are included in the abstract.

Reviewer 2 Report

Review of the paper

In the paper entitled “On The Solution of Thermal Buckling Problem of Moderately Thick Laminated Conical Shells Containing Carbon Nanotube Originating Layers” the thermal buckling of moderately thick laminated conical shells consisting of carbon nanotube (CNT). The overall presentation of the content is excellent, and the work is a great contribution to the field of buckling analysis of the CNT-reinforced laminated structure. The reviewer recommends this article for publication after the implementation of the comment indicated below.

1.     In this paper, theoretically it is assumed that the CNT is functionally graded along the thickness, also the alignment orientation of CNTs is changing with thickness coordinate. Is this possible to fabricate such kind of CNT reinforced structure?

2.     Where is the application of this kind of composite material?

3.     Page 4, “and their mounting on the subject make the solution more difficult to carry out.” Please rewrite the sentence by replacing the subject with an appropriate word.

4.     In table 1, what is “ (7, 12), (26,4), (27, 11), (28, 16), (29,19)”

5.     Same material properties of the CNT and PMMA are used in all layers. So, YScn(k) can be written as YScn only, it is not required to define the layer number (k). similarly, for all other properties of the CNT and PMMA matrix.

6.     Buckling results are compared with a simple isotropic truncated conical shell. But, the present model is developed for an advanced functionally graded laminated conical shell, so, a comparison study needs to show with functionally graded CNT composite shell structure.

7.     Title of table 3 is not conforming with the data of table 3.

Author Response

EXPLANATION TO REVIEWER 2:

First of all, we would like to thank the Reviewer 2 for his/her improving remarks and the time spent for them.

SUGGESTION 1: In this paper, theoretically it is assumed that the CNT is functionally graded along the thickness, also the alignment orientation of CNTs is changing with thickness coordinate. Is this possible to fabricate such kind of CNT reinforced structure?

EXPLANATION 1: Thanks. The CNT reinforced structural elements that used in this study have been manufactured since 2011. The production of these structural elements was carried out in the laboratory by Kwon et al. (2011).

Kwon H, Bradbury CR, Leparoux M. Fabrication of Functionally Graded Carbon Nanotube-Reinforced Aluminum Matrix Composite. Advanced Engineering Materials 2011,13(4), 325-329.

SUGGESTION 2:  Where is the application of this kind of composite material?

EXPLANATION 2:  Thanks. Nanocomposites are used more in the areas shown visually below (Kumar et al. 2016).

[5] Kumar, S., Reddy, M., Kumar, A., Devi, G.R.: Development and characterization of polymer – ceramic continuous fiber reinforced functionally graded composites for aerospace application. Aero Sci Technol, 26 (2013), pp. 185-191.

Today, among the nanocomposites formed by the reinforcement of the matrix with carbon nanotubes, polymer-based nanocomposites are more

popular (46% of publications) than metal (34% of publications) and ceramic-based (10% of publications) nanocomposites and are used more frequently in various industries [Dresselhaus et al., 1996, Duong et al, 2016).

[3] Dresselhaus, M.S., Dresselhaus, G., Eklund, P.C. :Science of Fullerenes and Carbon Nanotubes: Their Properties and Applications. Elsevier, Amsterdam, 1996.

[4]. Duong, H.M., Gong, F., Liu, P., Tran, T.Q.: Advanced fabrication and properties of aligned carbon nanotube composites: Experiments and modeling. In Carbon Nanotubes—Current Progress of Their Polymer Composites 2016; Berber, M.R., Hafez, I.H., Eds.; InTech: London, UK, 2016; ISBN 978-953-51-2470-2.

SUGGESTION 3:  Page 4. “ and their mounting on the subject make the solution more difficult to carry out” Please rewrite the sentence by replacing the subject with an appropriate word.

EXPLANATION 3: Thanks. This sentence has been modified as follows:

In addition, mathematical modeling of thermal loads and their inclusion in the main equations complicates the problem.

SUGGESTION 4: In table 1, what is (7,12), (26,4), (27,16), (29,19) .

EXPLANATION 4: Thanks. (7,12), (26,4), (27,16), (29,19)  are the wave numbers (m, n)  corresponding to the minimum values of the critical temperature.

SUGGESTION 5:  Same material properties of the CNT and PMMA are used in all layers. So, YScn(k) can be written as YScn only, it is not required to define the layer number (k). similarly, for all other properties of the CNT and PMMA matrix.

EXPLANATION 5: Thank you for your suggestion. In this case, there may be confusion between the multilayer and the single-layer shells. That's why it's good to keep it that way.

SUGGESTION 6:  Buckling results are compared with a simple isotropic truncated conical shell. But, the present model is developed for an advanced functionally graded laminated conical shell, so, a comparison study needs to show with functionally graded CNT composite shell structure.

EXPLANATION 6: Thanks. Since the main and reliable source for the buckling temperature of conical shells is the study of Lu and Chang (1957), a comparison is made with this study.

NOTE: It should be emphasized that the special case of the results obtained from this study (for the single-layer conical shell) coincides with the critical temperature values of single-layered nanocomposite conical shells and cylindrical shells (See, Avey et al. [40] and [41]).

In addition, in the study of Shen [51], the critical axial load of the nanocomposite cylindrical shell in the thermal environment is found numerically. A very special case of our results for buckling temperature (for single-layered cylindrical shell) also agrees with the numerical results of Shen [51], when the half-apex angle approaches zero.

 [40] Avey, M.; Sofiyev, A.H.; Kuruoglu, N. Influences of Elastic Foundations and Thermal Environments on the Thermoelastic Buckling of Nanocomposite Truncated Conical Shells. Acta Mech. 2022, 233 (2), 685-700.

[41] Avey, M.; Fantuzi, N.; Sofiyev, A.H. Mathematical Modeling and Analytical Solution of Thermoelastic Stability Problem of Functionally Graded Nanocomposite Cylinders within Different Theories. Mathematics 2022,10,1081.

[51]Shen, H.S. Postbuckling of Nanotube-Reinforced Composite Cylindrical Shells In Thermal Environments. Part I: Axially-loaded shells. Compos. Struct. 2011, 93, 2096–2108.

SUGGESTION 7:   Title of table 3 is not conforming with the data of table 3.

EXPLANATION 7: Table 3 and Table 4 are parts of the same table. Since the table is large, we divided it into Tables 3 and 4 so that it would fit on the pages. There is no problem with the title of Table 3 and it is compatible with the data.

Author Response

EXPLANATION TO REVIEWER 3:

First of all, we would like to thank the Reviewer 3 for his/her improving remarks and the time spent for them.

This paper presents thermal buckling of laminated truncated conical shells consisting of CNT originating layers subjected to uniform temperature rise. Authors have used the Donnell-type shell theory to derive the governing equations and Galerkin method to find the buckling temperature. This manuscript may be accepted for publication after revision addressing the following points:

SUGGESTION 1:  The objectives of the present study should be highlighted more in the last part of the introduction.

EXPLANATION 1: The aims of this study are highlighted in the last part of the introduction as follows:

The main purpose of this study is to solve the thermal buckling problem of moderately thick laminated conical shells consisting of CNT originating layers within the framework of shear deformation theory and to obtain a new analytical expression in the most general form. It is also to carry out detailed unique parametric studies using various shear stress functions to investigate the effects of CNT models, number and arrangement of layers on the buckling temperature, in comparison with the results in classical shell theory.

The manuscript is arranged as follows: The material properties of the nanocomposite layers and laminated conical shells reinforced with carbon nanotubes are modeled in Section 2. In Section 3, basic relations and equations are derived. The solution procedure in the framework of ST and CT is given in detail in Section 4. Numerical results and discussions are given in Section 5, followed by concluding remarks in Section 6.

SUGGESTION 2:  Authors have compared the present result with the following literature. Lu, S.Y.; Chang, L.K. Thermal Buckling of Conical Shells. AIAA J. 1957, 5(10),1877- 1882. Authors need to provide few more validations/comparisons of recent similar literatures.

EXPLANATION 2: Since the main and reliable source for the buckling temperature of conical shells is the study of Lu and Chang (1957), a comparison is made with this study.

NOTE: It should be emphasized that the special case of the results obtained from this study (for the single-layer conical shell) coincides with the critical temperature values of single-layered nanocomposite conical shells and cylindrical shells (See, Avey et al. [40] and [41]).

In addition, in the study of Shen [51], the critical axial load of the nanocomposite cylindrical shell in the thermal environment is found numerically. A very special case of our results for buckling temperature (for single-layered cylindrical shell) also agrees with the numerical results of Shen [51], when the half-apex angle approaches zero.

 [40] Avey, M.; Sofiyev, A.H.; Kuruoglu, N. Influences of Elastic Foundations and Thermal Environments on the Thermoelastic Buckling of Nanocomposite Truncated Conical Shells. Acta Mech. 2022, 233 (2), 685-700.

[41] Avey, M.; Fantuzi, N.; Sofiyev, A.H. Mathematical Modeling and Analytical Solution of Thermoelastic Stability Problem of Functionally Graded Nanocomposite Cylinders within Different Theories. Mathematics 2022,10,1081.

[51]Shen, H.S. Postbuckling of Nanotube-Reinforced Composite Cylindrical Shells In Thermal Environments. Part I: Axially-loaded shells. Compos. Struct. 2011, 93, 2096–2108.

SUGGESTION 3:  Please provide the reference in the Table 2.

EXPLANATION 3: Thanks. The reference Shen [51] is added in Table 2.

SUGGESTION 4: When the UBT values for all arrays of laminated conical shells are compared, almost the same results are obtained for the Par-and Cos-Hyp-TSS functions. Why?

EXPLANATION 4: Thanks. You are right, when Cos-Hyp- and Par-transverse shear stress  (Cos-Hyp-TSS and Par-TSS) functions are used as shear stress functions, a value set is given because the critical temperature values ​​differ in the  4th and fifth digits of the comma (see Table 1* for Reviewers). Mathematically, this means that since the numerical values ​​of the integrals in terms of these functions are almost the same, the values ​​of the buckling temperature are also the same. Therefore, it turns out that approximately the same results are obtained when Cos-Hyp-TSS and Par-TSS functions are used.

Table 1* (for Reviewers) Variations of UBT for single-layer and laminated cones with CNT -shaped layers for Par- and Cos-Hyp-SS functions for  = 10°

= 10°

 (ncr) for Par (m=1)

Arrangement  of  layers

U

V

ST

CT

ST

CT

(0o)

0.7886089971 (5)

1.375054951 (4)

0.6623625174 (4)

0.9721811089 (4)

(0o/90o/0o)

1.015253916 (5)

1.712211543 (5)

0.8150226251 (5)

1.060832094 (5)

(90o/0o/90o)

0.5615988629 (3)

0.5844530616 (3)

0.5342772206 (4)

0.5781959494 (4)

(0o/90o/0o/90o)

0.8795123466 (4)

1.085330656 (4)

0.8024069521 (4)

0.9760816848 (4)

(0o/90o/90o/0o)

1.165394406 (5)

1.837398691 (4)

0.888920668 (4)

1.037713209 (4)

(90o/0o/0o/90o)

0.5991216424 (3)

0.6451522084 (3)

0.5674921594 (4)

0.6188677499 (4)

Arrangement  of  layers

O

X

ST

CT

ST

CT

(0o)

0.5526068956 (4)

0.7537447114 (4)

0.9320270558 (5)

1.991222944 (4)

(0o/90o/0o)

0.8944756978 (5)

1.173700302 (5)

0.9767367035 (5)

2.047765193 (5)

(90o/0o/90o)

0.3908647278 (4)

0.4162628424 (4)

0.8033860213 (3)

0.8511516793 (3)

(0o/90o/0o/90o)

0.6552096035 (4)

0.7580773693 (4)

0.8759402171 (4)

1.052976572 (4)

(0o/90o/90o/0o)

1.036114187 (4)

1.258322092 (4)

1.081848047 (5)

2.151361554 (4)

(90o/0o/0o/90o)

0.4288874994 (4)

0.475280073 (4)

0.8279833725 (3)

0.8949272732 (3)

Cos-ST (m=1)

Arrangement  of  layers

U

V

ST

CT

ST

CT

(0o)

0.7889998522 (5)

1.375054951 (4)

0.6626227572 (4)

0.9721811089 (4)

(0o/90o/0o)

1.015683028 (5)

1.712211543 (5)

0.8152719186 (5)

1.060832094 (5)

(90o/0o/90o)

0.5616339913 (3)

0.5844530616 (3)

0.5343308452 (4)

0.5781959494 (4)

(0o/90o/0o/90o)

0.8797749139 (4)

1.085330656 (4)

0.8026408237 (4)

0.9760816848 (4)

(0o/90o/90o/0o)

1.165765822 (5)

1.837398691 (4)

0.8890898234 (4)

1.037713209 (4)

(90o/0o/0o/90o)

0.5992169554 (3)

0.6451522084 (3)

0.5675930639 (4)

0.6188677499 (4)

Arrangement  of  layers

O

X

ST

CT

ST

CT

(0o)

0.5528578285 (4)

0.7537447114 (4)

0.9325621293 (5)

1.991222944 (4)

(0o/90o/0o)

0.894787302 (5)

1.173700302 (5)

0.9772611732 (5)

2.047765193 (5)

(90o/0o/90o)

0.390910039 (4)

0.4162628424 (4)

0.8034381476 (3)

0.8511516793 (3)

(0o/90o/0o/90o)

0.6553527079 (4)

0.7580773693 (4)

0.8762123912 (4)

1.052976572 (4)

(0o/90o/90o/0o)

1.036318973 (4)

1.258322092 (4)

1.08233884 (5)

2.151361554 (4)

(90o/0o/0o/90o)

0.4289859943 (4)

0.475280073 (4)

0.82809336 (3)

0.8949272732 (3)

NOTE 1: This shows that using either of the Cos-Hyp-TSS and Par-TSS functions is more reliable than using the U-TSS function.

This is not contrary to the general application principles of shear deformation theory. It should be emphasized that the obtained formulas are derived for general shear stress functions fi(z), (i=1,2). In numerical analysis, detailed analyzes were made by choosing three different shear stress functions.

NOTE 2: Other approximation functions are also available in the literature:

Apart from abovementioned models, there are some models

SUGGESTIONS 4 and 5:  Table 3. Variations of UBT for single-layer and laminated cones with CNT -shaped layers for Par- and Cos-Hyp-SS functions. What basis the number of layers and stacking sequence are chosen. Whether the same trend will be obtained for angle ply laminates.

  • Table 4. Variations of UBT for single-layer and laminated cones with CNT -shaped layers for the U- TSS function. Please explain the reasons for the results.

EXPLANATIONS 4 and 5: Thanks. Table 3 and Table 4 are parts of the same table. Since the table is large, we divided it into Tables 3 and 4 so that it would fit on the pages. Angle-ply laminates were not studied in this study. This is a different matter, and all of the analytical formulas for critical temperature will vary, from fundamental relations to fundamental equations. It is an interesting topic that can be done in the future.

When we compare all laminated and (0o)- single-layer conical shells for the Par- TSS (or Cos-Hyp- TSS) function, the biggest differences between UBT values are found in the U-, V-, O-shaped (0o/90o/0o/90o)-array cone when , which are 47.66%, 34.29%, 88.71%, respectively, while in the X-pattern, it occurs at (90o/0o/90o)-array cone is (-23.7%) when  (Table 3). When comparing all laminated and (0o)- single-layer conical shells for the U- TSS  function, the biggest differences between UBT values are obtained in the U-, V-, O-shaped (0o/90o/0o/90o)-array conical shell, as , which are 44.93%, 30.81%, 83.36%, respectively, while in the X-pattern, it occurs at (90o/0o/90o)-array conical shell is (-30.84%), as  (Table 4). As can be seen from Tables 3 and 4, the influence of the arrangement and number of layers on the buckling temperature is reduced when using U-TSS compared to Par-TSS (or Cos-Hyp-TSS) functions in U-, V- and O-shaped conical shells, while this effect is more pronounced in nanocomposite conical shells with an X-shaped pattern.

SUGGESTION 6: All the tables and figures must be replotted for uniformity of the fonts and clarity.

EXPLANATION 6: All tables and figures have been redrawn and drawn for uniformity and clarity of fonts.

Fig. 4. Variations of UBT for (90o/0o/90o)- and (0o/90o/0o)-array laminated cones with U- and X shaped layers for Cos-Hyp-TSS function and within CT against

Fig. 5. Variations of UBT for (0o/90o/0o/90o)- and (90o/0o/0o/90o)-array laminated cones with U- and V -shaped layers for Cos-Hyp-TSS function and within CT against  

Fig. 6. Variations of UBT for (0o)-single-layer and (0o/90o/0o/90o)-array laminated cones with U- and X -shaped layers for Cos-Hyp-TSS function and within CT against  

SUGGESTION 7:  There are parts of the paper which are incomprehensible so the English has to be revised. Please check for grammatical and typographical errors. Generally, the authors should do a thorough proof read again.

EXPLANATION 7: We have made a careful check throughout the whole manuscript, and tried to identify and eliminate all the grammatical mistakes.

SUGGESTION 8:  Ensure that the references are strictly as per the format of the publisher.

EXPLANATION 8: Rechecked to make sure the references match the publisher's format.

Reviewer 4 Report

The paper studied the thermal buckling of laminated truncated conical shells composed of CNT originating layers within STs. In this study the Donnell theory is applied to derive the basic equations and then the Galerkin method is applied to the basic equations to find expression for UBT of laminated truncated conical shells composed of CNT originating layers within ST and CT. The influences of changes of CNT models, the arrangement and number of the layers on the UBT using different shear stress functions are examined. The submitted paper could be of interest to journal readers, however, many updates/corrections are needed:

1)          The buckling phenomenon is a vital problem to be studied during the analysis and design of several fields of engineering applications. It can be seen from the literature that considerable efforts are made for the analysis of buckling of structures plates/shells. Different theories have been adopted to analysis the buckling behavior of structures. A brief literature review should be addressed to cover the different method used to studies the buckling analysis. The following works have potential to be considered in that part of literature review. A) Buckling analysis of carbon nanotube reinforced FG shells using an efficient solid-shell element based on a modified FSDT Thin-Walled Struct. Vol.144, 106254, 2019 B) “A modified first shear deformation theory for three-dimensional thermal post-buckling analysis of FGM plates.” Meccanica 57, 337–353 (2022. C) “Three-dimensional thermal buckling analysis of functionally graded material structures using a modified FSDT-based solid-shell element.” International Journal of Pressure Vessels and Piping 194 (2021) 104547. Based on the introduced works, this issue should be addressed in the manuscript in detail.

2)          The main novelty of this work should be clarified in the last paragraph of introduction.

3)          The main physical application of this work should be added to the paper.

4)          Is a linear stability analysis is valid for all the shell-type structures under thermal loading? This issue should be discussed with specific focus on the conical shell. 

5)          In the buckling analysis, the possibility for mode shifting (changing the first mode of buckling) due to the temperature-dependency or the nanocomposite layers materials (volume fractions of CNTs) should be investigated in detail for both the shell cases? It is recommended to provide tabulated results to address this comment.

6)          Conclusion section must be extended in a few words via main finding and advantages of the methodology.

Author Response

EXPLANATION TO REVIEWER 4:

First of all, we would like to thank the Reviewer 3 for his/her improving remarks and the time spent for them.

The paper studied the thermal buckling of laminated truncated conical shells composed of CNT originating layers within STs. In this study the Donnell theory is applied to derive the basic equations and then the Galerkin method is applied to the basic equations to find expression for UBT of laminated truncated conical shells composed of CNT originating layers within ST and CT. The influences of changes of CNT models, the arrangement and number of the layers on the UBT using different shear stress functions are examined. The submitted paper could be of interest to journal readers, however, many updates/corrections are needed:

SUGGESTION 1: The buckling phenomenon is a vital problem to be studied during the analysis and design of several fields of engineering applications. It can be seen from the literature that considerable efforts are made for the analysis of buckling of structures plates/shells. Different theories have been adopted to analysis the buckling behavior of structures. A brief literature review should be addressed to cover the different method used to studies the buckling analysis. The following works have potential to be considered in that part of literature review.

  1. A)Hajlaoui, E.Chebbi, F.Dammak Buckling analysis of carbon nanotube reinforced FG shells using an efficient solid-shell element based on a modified FSDTThin-Walled Struct. Vol.144, 106254, 2019
  2. B) A. Hajlaoui & F. Dammak A modified first shear deformation theory for three-dimensional thermal post-buckling analysis of FGM plates.” Meccanica 57, 337–353 (2022.
  3. C) “Three-dimensional thermal buckling analysis of functionally graded material structures using a modified FSDT-based solid-shell element.” International Journal of Pressure Vessels and Piping 194 (2021) 104547.

Based on the introduced works, this issue should be addressed in the manuscript in detail.

EXPLANATION 1: Thanks. Your suggestion is implemented as follows:

Single-layer heterogeneous composites reinforced with carbon nanotubes, which have robust heat resistance and high strength, are frequently used in the spacecraft and aerospace industries because of their good performance in very high temperature conditions, and the number of studies on their thermal buckling behavior is increasing rapidly [26-42].

The following articles were reviewed and used in the manuscript as they are related to the subject of the study.

[34] Hajlaoui, A.; Chebbi E.; Dammak, F. Buckling Analysis of Carbon Nanotube Reinforced FG Shells Using an Efficient Solid-Shell Element Based on a Modified FSDT. Thin-Walled Struct. 2019, 144, 106254.

[35] Hajlaoui, A.; Chebbi E.; Dammak, F. Three-Dimensional Thermal Buckling Analysis of Functionally Graded Material Structures Using a Modified FSDT-Based Solid-Shell Element. Int. J. Press. Vessel. Pip 2021,194,104547.

[42] Hajlaoui, A.; Dammak, F. A Modified First Shear Deformation Theory for Three-Dimensional Thermal Post-Buckling Analysis of FGM Plates. Meccanica 2022, 57, 337–353.

SUGGESTION 2: The main novelty of this work should be clarified in the last paragraph of introduction.

EXPLANATION 2: The following additions are made to the end of the introduction in the revised manuscript:

The main purpose of this study is to solve the thermal buckling problem of moderately thick laminated conical shells consisting of CNT originating layers within the framework of shear deformation theory and to obtain a new analytical expression in the most general form. It is also to carry out detailed unique parametric studies using various shear stress functions to investigate the effects of CNT models, number and arrangement of layers on the buckling temperature, in comparison with the results in classical shell theory.

SUGGESTION 3: The main physical application of this work should be added to the paper.

EXPLANATION 3: The following statement is added to the result section:

Since laminated heterogeneous nanocomposite conical shells reinforced with carbon nanotubes with robust heat resistance and high strength are frequently used in the nuclear and aerospace industries exposed to very high temperatures, the results obtained in this research on their thermal buckling behavior should be considered during design.

SUGGESTION 4: Is a linear stability analysis is valid for all the shell-type structures under thermal loading? This issue should be discussed with specific focus on the conical shell.

EXPLANATION 4: Thanks. It is not possible to obtain results for buckling temperature for all shell types from the buckling temperature expression of conical shells. However, numerical magnitudes and analytical expressions for the buckling temperature of cylindrical shells based on the ST and CT can be specially obtained when the half apex angle approaches zero (see, Ref. [40]).

SUGGESTION 5: In the buckling analysis, the possibility for mode shifting (changing the first mode of buckling) due to the temperature-dependency or the nanocomposite layers materials (volume fractions of CNTs) should be investigated in detail for both the shell cases? It is recommended to provide tabulated results to address this comment.

EXPLANATION 5: Thanks. Two different volume fractions are used in this study. In Tables 3 and 4 is used and in Table 5 and Figures 4-6 is used . Here, the suggestion of the other referee was taken into account, also.

SUGGESTION 6: Conclusion section must be extended in a few words via main finding and advantages of the methodology.

EXPLANATION 6: Thanks. The conclusion part has been modified as follows:

The thermal buckling of laminated truncated conical shells composed of CNT originating layers within STs is studied. The modified Donnell type shell theory is applied to derive the basic equations and then the Galerkin method is applied to the basic equations to find expression for UBT of laminated truncated conical shells composed of CNT originating layers within ST and CT. Four types of single-walled carbon nanotube distribution across the thickness of the layers are considered, namely, uniform and functionally graded. The Par-, Cos-Hyp- and U-transverse shear stress functions are used in the analysis. The influences of change of CNT models, the arrangement and number of the layers on the UBT using different shear stress functions are examined.

……………………………………………………….

Since laminated heterogeneous nanocomposite conical shells reinforced with carbon nanotubes with robust heat resistance and high strength are frequently used in modern aerospace and rocket technology, shipbuilding, energy and chemical engineering, and other fields exposed to very high temperatures, the results obtained in this research on their thermal buckling behavior should be considered during desig

Round 2

Reviewer 2 Report

paper can accept

Reviewer 4 Report

The revised paper could be of interest to journal readers, however, and I recommend that it be accepted in its present form.